# PATH COMPLEX MESSAGE PASSING FOR MOLECULAR PROPERTY PREDICTION

## ABSTRACT

Geometric deep learning (GDL) has demonstrated enormous power in molecular data analysis. However, GDL faces challenges in achieving high efficiency and expressivity in molecular representations when high-order terms of the atomic force fields are not sufficiently learned. In this work, we introduce message passing on path complexes, called the Path Complex Message Passing, for molecular prediction. Path complexes represent the geometry of paths and can model the chemical and non-chemical interactions of atoms in a molecule across various dimensions. Our model defines messages on path complexes and employs neural message passing to learn simplex features, enabling feature communication within and between different dimensions. Since messages on high-order and low-order path complexes reflect different aspects of molecular energy, they are updated sequentially according to their order. The higher the order of the path complex, the richer the information it contains, and the higher its priority during inference. It can thus characterize various types of molecular interactions specified in molecular dynamics (MD) force fields. Our model has been extensively validated on benchmark datasets and achieves state-of-the-art results. The code is available at `https://anonymous.4open.science/r/Path-Complex-Neural-Network-32D6`

## 1 INTRODUCTION

Accurate prediction of molecular properties is crucial in fields such as drug design Zhang et al. (2017); Chen et al. (2018); Mak & Pichika (2019); Chan et al. (2019), biology Townshend et al. (2021); Jamasb et al. (2022), chemistry Qiao et al. (2022), and materials science Vlassis et al. (2020). Geometric Deep Learning (GDL) has demonstrated significant potential in molecular sciences, leading to a surge in studies employing GDL models for effective molecular representation learning Bronstein et al. (2017); Atz et al. (2021); Ingraham et al. (2023). Among the three types of representations used in GDL models—topological, geometric, and functional—the molecular graph has become the most popular due to its simplicity, flexibility, and efficiency Wieder et al. (2020); Yu & Gao (2022); Atz et al. (2021); Li et al. (2022); Wang et al. (2022b). However, relying solely on graph representations fails to capture the many-body interactions inherent in complex systems, thereby limiting the expressiveness and predictive power of this approach Bodnar et al. (2021b). This paper develops a path complex-based neural message passing for molecule prediction, where the molecular energy of force field can be well represented.

In Graph Neural Networks (GNNs), the molecular graph is typically constructed based on covalent bonds. Node features are usually derived from atomic properties and are updated by aggregating information from neighboring nodes Huang et al. (2020); Shindo & Matsumoto (2019); Shui & Karypis (2020a); Schütt et al. (2017); Unke & Meuwly (2019). To enhance GNN performance, researchers have proposed several approaches. One major strategy is to design more complex molecular graphs that incorporate non-covalent interactions. The most common method involves introducing edges between any two atoms within a specified cutoff distance, effectively capturing non-covalent interactions. Additionally, molecule-based line graph models have been developed, where nodes represent atomic bonds and edges represent bond angles Choudhary & DeCost (2021).

The second approach focuses on incorporating global physical features and local geometric information into GNN models. Global physical attributes such as temperature, pressure, and entropy

have been added to GNN architectures to better characterize molecular states and environments, as demonstrated in MEGNet Chen et al. (2019) and SphereNet Liu et al. (2022). Local geometric features—particularly bond lengths, bond angles Schütt et al. (2018); Flam-Shepherd et al. (2021), dihedral angles Wang et al. (2022a), and torsion angles, which are crucial to molecular properties—have been extensively considered in models such as DimeNet Gasteiger et al. (2020), GemNet Gasteiger et al. (2021), ALIGNN Choudhary & DeCost (2021), and GEM Fang et al. (2022).

Another approach involves designing efficient message-passing modules for invariant features, equivariant properties, and higher-order tensors. The expressivity of GNNs is closely related to the message-passing mechanisms used in layers that process invariant, equivariant, or higher-order tensor features. These three approaches are often synergistically integrated to enhance model performance.

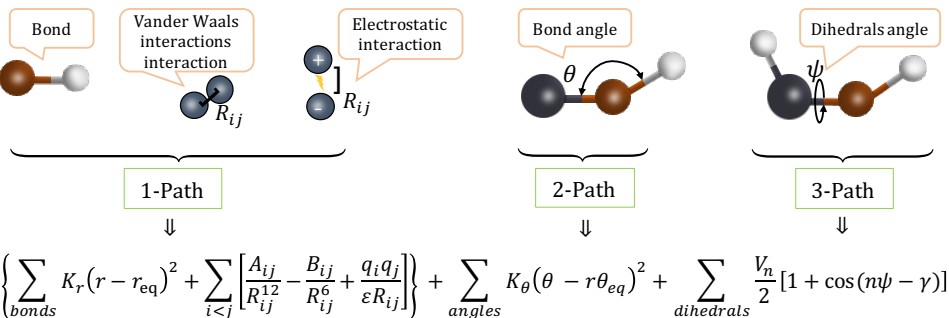

$$\left\{\sum_{bonds} K_r\left(r - r_{\text{eq}}\right)^2 + \sum_{i<j}\left[\frac{A_{ij}}{R_{ij}^{12}} - \frac{B_{ij}}{R_{ij}^6} + \frac{q_i q_j}{\varepsilon R_{ij}}\right]\right\} + \sum_{angles} K_\theta\left(\theta - r\theta_{eq}\right)^2 + \sum_{dihedrals} \frac{V_n}{2}\left[1 + \cos\left(n\psi - \gamma\right)\right]$$

Figure 1: Terms of the approximate equation to molecular dynamics force field correspond to path complices of order one to three, which have been used in path complex message passing.

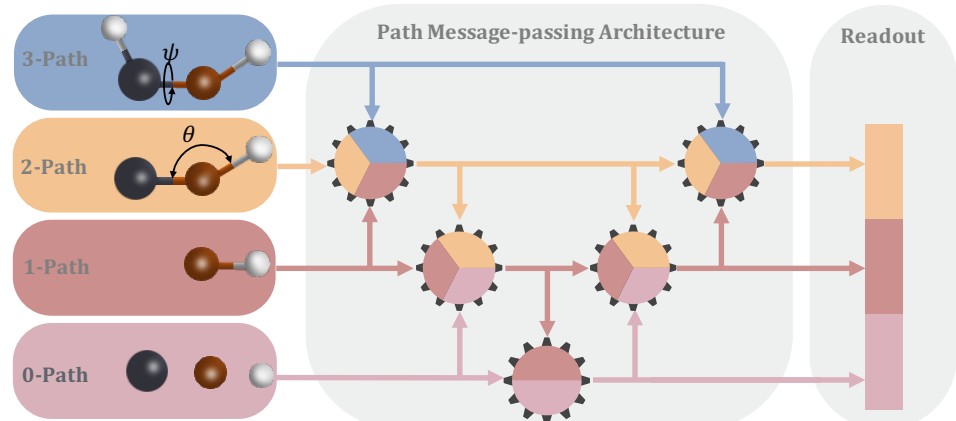

Figure 2: The architecture of PCMP utilizing path complexes up to order 3 is depicted. At each layer $l$, each path complex message of a given order updates its features using messages from path complexes of the same order and adjacent orders from the previous layer $l - 1$. Higher-order path complex messages are updated before lower-order ones because the former encompass the paths of the latter. Additionally, the interplay between high-order and low-order path complexes is learned through message passing.

In this work, we develop path complex-based molecular representation and path complex message passing (PCNN) model for molecular property analysis. PCNN is a neural message passing Gilmer et al. (2017) on path complices. A path is a sequence of points, and a path complex is a subset of all possible paths. In the context of a molecule or molecular graph, a path corresponds to the geometry defined by chemical or non-chemical bonds. Our path complexes are specifically designed — based on molecular graphs that include both covalent and non-covalent bonds — to characterize different types of energy specified in molecular dynamics (MD) force fields, as shown in Figure 1. The MD potential energy Mayo et al. (1990); González (2011); Leach (2001) comprises bond terms ($E_B$,

two-body interactions), bond-angle terms ($E_A$, three-body interactions), and dihedral-angle terms ($E_T$, four-body interactions), which are effectively characterized by our 1-path, 2-path, and 3-path features, respectively.

Each path complex is assigned a message. Similar to classical neural message passing on graphs Gilmer et al. (2017) and simplicial complexes Bodnar et al. (2021a;b), the propagation of messages in a path complex is influenced by the messages of its "adjacent" path complexes at different orders. A higher-order path complex contains longer paths and includes the shorter paths of lower-order path complexes. Therefore, we need to update messages according to the order of the path complexes: messages in higher-order path complexes have priority in being updated. However, since path complexes of adjacent orders are interconnected, we incorporate interactions between higher-order and lower-order path complex information during message passing. Specifically, the higher-order message first updates the lower-order one, and then the updated lower-order message exerts a reverse effect on the higher-order information. Figure 2 illustrates a neural message passing process among path complexes of different orders designed based on this principle.

PCNN thus enables information passing between path complex features, using the aggregated information to predict molecular properties. Testing on benchmark datasets demonstrates promising performance. Our contributions are as follows:

1. We have developed a path complex-based molecular representation that explicitly characterizes different terms in the molecular dynamics (MD) force field for molecular property prediction.

2. We propose constructing path complexes using the unique topologies of graphs, simplicial complexes, and hypergraphs. This method, based on path complex message passing, enables the exchange of information across multiple dimensions.

3. Our PCMP model has been rigorously tested and validated on benchmark molecular tasks, consistently achieving state-of-the-art results.

## 2 RELATED WORK

**Graph Neural Networks for Molecular Property Prediction**     Graph neural network models have played an pivotal role in molecular data analysis. Traditional GNN models represent molecules as the de factor covalent-bond-based molecular graphs, and use major GNN architectures, such as GIN Xu et al. (2018), GAT Velickovic et al. (2017), GCN Kipf & Welling (2016a), SGCN Danel et al. (2020) and GTtransformer Rong et al. (2020), to learn molecular properties Yang et al. (2019); Xiong et al. (2019); Choudhary & DeCost (2021); Fang et al. (2022). With the importance of non-covalent bonds, cutoff-distance-based molecular graph representations have been widely employed in GNN models, such as DimeNet Gasteiger et al. (2020), HMGNN Shui & Karypis (2020b), GeoGNN Fang et al. (2022), Mol-GDL Shen et al. (2023), etc. Further, higher-order interactions (beyond pair-wise forces) has been explicitly incorporated into GNN models, including ALIGNN Choudhary & DeCost (2021), GEM Fang et al. (2022), DimeNet Gasteiger et al. (2020), GemNet Gasteiger et al. (2021), etc, by the consideration of bond angles, dihedral angles, torsion angles, and other local geometric information. In particular, these higher-order terms can be directly related to MD force field information Halgren (1996); Choudhary et al. (2018). Finally, pre-training process has been adopted to further improve the accuracy of GNN models, such as N-Gram Liu et al. (2019), PretrainGNN Hu et al. (2019), GEM Fang et al. (2022), MolCLR Wang et al. (2022b), DMP Zhu et al. (2023), etc.

**Topological Deep Learning (TDL)**     Topological Deep Learning (TDL) Hajij et al. (2022); Bodnar (2022) leverages novel topological tools to characterize data with complicated higher-order structures. Different from graph-based data representation, TDL uses topological representations from algebraic topology, including simplicial complexes Bodnar (2022); Schaub et al. (2022), cell complexes Hajij et al. (2020); Roddenberry et al. (2022); Giusti et al. (2023), sheaves Hansen & Ghrist (2019); Bodnar et al. (2021b), hypergraphs Feng et al. (2019); Kim et al. (2020); Bai et al. (2021), and combinatorial complexes Hajij et al. (2022) to model not only pair-wise interactions (as in graphs), but also higher-order interactions among three or more elements. In fact, these algebraic topology-based molecular representations have already achieved great success in molecular data

analysis, including protein flexibility and dynamic analysis Xia & Wei (2014); Sverrisson et al. (2021), drug design Cang & Wei (2017), virus analysis Chen et al. (2022), materials property analysis Reiser et al. (2022); Townsend et al. (2020). Further, TDL uses a generalized message-passing mechanism thus enables the communication of information from simplices of different dimensions. In contrast to GNNs, where information is passing among nodes or edges, TDL allows information to propagate through any neighborhood relation Roddenberry et al. (2021).

Recently, path complex and its related models, including path homology Grigor'yan et al. (2018), persistent path homology Chowdhury & Mémoli (2018); Liu et al. (2023); Chen et al. (2023), path Laplacian Wang & Wei (2023), a special path-complex-based topological message passing model Truong & Chin (2024) has been developed and demonstrated great potential for the analysis of molecular structures.

**Geometric Deep Learning and Molecular Representation**   Generally speaking, molecules in GDL models are characterized by three types of molecular representations, including topological representations (such as molecular graphs), geometric representation (such as molecular surfaces), and function representation (such as molecular density). Deep learning models including (3D) convolutional neural networks, graph neural networks (GNNs), recurrent neural networks, and others, have been constructed based on these representations Wieder et al. (2020); Yu & Gao (2022); Atz et al. (2021); Li et al. (2022); Wang et al. (2022b). With its simplicity, flexibility and efficiency, molecular graphs are the most popular of various types of GNN models have been proposed, including graph recurrent neural networks (GraphRNN) You et al. (2018), graph convolutional networks (GCN) Welling & Kipf (2016), graph autoencoders Kipf & Welling (2016b), graph transformers Yun et al. (2019), etc. These GNN models have been widely used in molecular data analysis.

## 3   PATH COMPLEX MESSAGE PASSING

Path complex was originally developed on directed graph (or digraph) and set, by Grigoryan, Lin, Muranov and Yau in 2012 Grigor'yan et al. (2012). They also proposed a new homology theory for path complex, called path homology, and use it to explore topological invariant information of digraphs Grigor'yan et al. (2014). Mathematically, path homology provides a novel framework to systematically explore intrinsic topological information of more general structures Grigor'yan et al. (2019); Grigor'yan et al. (2020). Details of path complex and path homology can be found in the Appendix B.

Here we propose a generalized way to construct path complex based on undirected graph, simplicial complex, and hypergraph. On undirected graph, we propose graph collapse and expansion operations, and use them to systematically study graph isomorphism by their path complex homology groups. We found that the path complex homology is a graph weak isomorphism invariant. For simplicial complex and hypergraph, we propose simplex- and hyperedge- based path complex.

### 3.1   GENERALIZED PATH COMPLEX

**Path complex for undirected-graph**   Firstly, we give the construction of path complex for undirected graphs. Secondly, we introduce the graph weak isomorphism and related mathematical properties. Finally, we states the weak isomorphism invariance of path complex homology for graphs.

**Definition 3.1** (Path). Given a simple undirected graph $G = (V, E)$ over the verset set $V$, an $n$-path $\sigma_n$ of $G$ is defined as any sequence of $n + 1$ vertices $v_0 v_1 \cdots v_n (v_i \in V)$ such that every two vertices are distinct and every two adjacent vertices form an edge.

Note that for each $n$-path $\sigma_n = v_0 v_1 \cdots v_n$, $\sigma'_n = v_n \cdots v_1 v_0$ is also an $n$-path, we identify these two paths as the same one. For an $n$-path $\sigma_n = v_0 \cdots v_n$, the $(n-1)$-paths by removing the first or last vertex, denoted by $\partial^L_{\sigma_n}$ and $\partial^R_{\sigma_n}$ respectively, are called the faces of $\sigma_n$. Two $n$-paths are neighbors if they are faces of a common $(n + 1)$-path. Let $\mathcal{N}(\sigma_n)$ be the set of neighbors of $\sigma_n$.

**Definition 3.2** (Path complex from undirected graphs). Given a simple undirected graph $G = (V, E)$, all paths of $G$ form a path complex $P_G$. We call $P_G$ the path complex derived from $G$.

**Theorem 3.3** (Path Complex Invariance). *The graph neural network is invariant to the permutation of the simplexes in the path complex $P_G$, meaning the output of the network remains unchanged under any permutation $\pi$ of the vertices.*

**Definition 3.4** (Graph collapse and expansion ). Given a graph $G = (V, E)$, take an edge $(v_1, v_2) \in E$ such that $deg(v_1) = 1$. Let $V' = V \backslash \{v_1\}$, $E' = E \backslash \{(v_1, v_2)\}$, then $G' = (V', E')$ is a new graph. We say that $G'$ is derived from $G$ by a graph collapse and $G$ is derived from $G'$ by a graph expansion.

**Definition 3.5** (Weak isomorphism). Given two graphs $G_1, G_2$, $G_1$ and $G_2$ are called weak isomorphic if $G_1$ can be derive from $G_2$ by a sequence of graph collapse and expansion operations.

It can be seen that two graphs are weak isomorphic if they are isomorphic.

**Theorem 3.6.** *If two graphs $G_1$ and $G_2$ are weak isomorphic, then, 1) The number of connected components of $G_1$ and $G_2$ are same; 2) The number of cycles of $G_1$ and $G_2$ are same.*

**Theorem 3.7.** *Given two graphs $G_1, G_2$, let $P_{G_1}, P_{G_2}$ be the path complexs derived from $G_1$ and $G_2$ respectively. If $G_1$ and $G_2$ are weak isomorphic, then*

$$\mathrm{H}_k(P_{G_1}) \cong \mathrm{H}_k(P_{G_2}) \ (k \geqslant 0)$$

Theorem 3.7 means the path complex homology is a graph weak isomorphism invariant. Consequently, for two graphs $G_1$ and $G_2$, if there exists $k$ such that $\mathrm{H}_k(P_{G_1}) \not\cong \mathrm{H}_k(P_{G_2})$, then $G_1$ and $G_2$ are not weak isomorphic and not isomorphic.

The profound theoretical relationship between the Weisfeiler-Lehman (WL) graph isomorphism test and message-passing graph neural networks (GNNs) has been extensively documented Xu et al. (2018).

**Definition 3.8** (PWL). The steps of general PWL are as follows:

1. Given a path complex $P$, all the paths of $P$ are initialized with the same color.

2. For the color $c_\sigma^t$ of path $\sigma$ at iteration $t$, the color $c_\sigma^{t+1}$ of $\sigma$ at the next iteration is computed by perfectly hashing the color multi-set of the neighbors of $\sigma$.

3. The algorithm stops once a stable coloring is reached. Two path complexes are considered non-isomorphic if their color histograms are different at some dimensions.

**Theorem 3.9.** *PWL is strictly more powerful than WL.*

Figure 8 shows two graphs that cannot be distinguished by the WL test, but their derived path complexes can be distinguished by PWL.

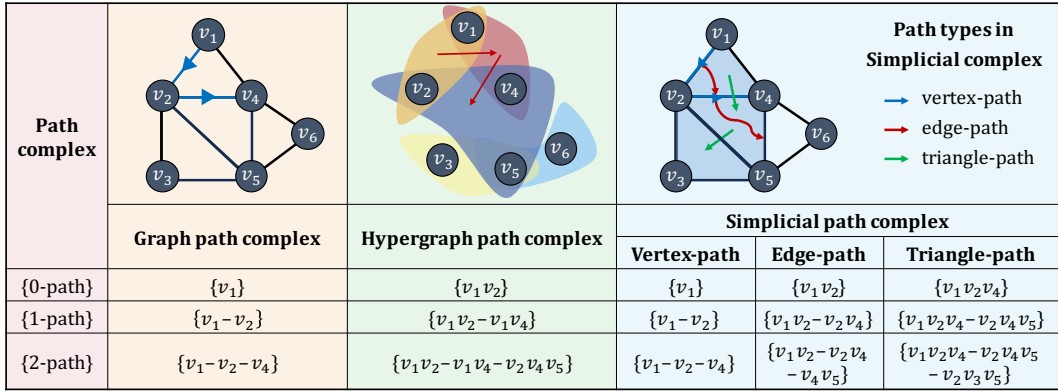

| Path complex | Graph path complex | Hypergraph path complex | Simplicial path complex | | |
|---|---|---|---|---|---|
| | | | Vertex-path | Edge-path | Triangle-path |
| {0-path} | $\{v_1\}$ | $\{v_1 v_2\}$ | $\{v_1\}$ | $\{v_1 v_2\}$ | $\{v_1 v_2 v_4\}$ |
| {1-path} | $\{v_1 - v_2\}$ | $\{v_1 v_2 - v_1 v_4\}$ | $\{v_1 - v_2\}$ | $\{v_1 v_2 - v_2 v_4\}$ | $\{v_1 v_2 v_4 - v_2 v_4 v_5\}$ |
| {2-path} | $\{v_1 - v_2 - v_4\}$ | $\{v_1 v_2 - v_1 v_4 - v_2 v_4 v_5\}$ | $\{v_1 - v_2 - v_4\}$ | $\{v_1 v_2 - v_2 v_4 - v_4 v_5\}$ | $\{v_1 v_2 v_4 - v_2 v_4 v_5 - v_2 v_3 v_5\}$ |

Figure 3: The path complexes of graphs, simplicial complexes, and hypergraphs. The table lists the 0-paths, 1-paths, and 2-paths, where the red arrows indicate the selected 2-paths. Specifically, for simplicial complexes, we enumerate the path complexes from vertexes (0-simplices), edges (1-simplices) and triangles (2-simplices), respectively.

**Path Complex for simplicial complex and hypergraph** Generally speaking, a path complex is a set of paths that is closed under the removing of the first or last vertices of each path. So we can also construct path complex from simplicial complex and hypergraph by defining simplex-paths and

hyperedge-paths. Various kinds of paths can be defined by considering the lower adjacent, upper adjacent, face and coface relations among simplices and hyperedges. Figure 3 shows examples of path complexes constructed from graph, simplicial complex and hypergraph. Details can be found in Appendix B.1.

### 3.2 MOLECULAR PATH COMPLEX REPRESENTATION AND PATH FEATURES

**Molecular Path Complex Representation** Currently, covalent-bond molecular graphs serve as the standard for molecular topological representations. These graphs underpin the molecular force fields used in molecular dynamics simulations, incorporating terms for both covalent bonds—such as bond lengths, angles, and dihedral angles—and non-covalent interactions like electrostatic and van der Waals forces. To enhance molecular representations with comprehensive force field data, we introduce the molecular path complex. This model utilizes path simplices across different dimensions to distinctly represent both covalent and non-covalent bond terms. As depicted in Figure 5 (in Appendix A.1), the $C_2H_6O$ molecule is illustrated alongside its corresponding path simplices. Specifically, our 1-path simplex captures bond lengths, the 2-path simplex details bond angles, and the 3-path simplex reflects dihedral angles.

**Path Features** Our path (simplex) features are meticulously designed to encapsulate the various atomic properties and interactions detailed in molecular dynamics (MD) force fields. Specifically, our 0-path features—comprising atomic number, radius, and electronegativity—are derived using Rdkit, akin to the approach in CGCNN Xie & Grossman (2018). Table 7 (in Appendix A.1) presents a comprehensive listing of our 1-path, 2-path, and 3-path features. Importantly, our model employs detailed local geometric properties of the path complex as path features. This method allows us to explicitly learn covalent bond terms defined in the MD force fields, while also implicitly capturing non-bond interactions.

### 3.3 MOLECULAR PCMP MODEL

Path Complex Message Passing (PCMP) introduces a novel method for message passing in graphs by leveraging path complexes, which are composed of paths of varying lengths. In contrast to traditional graph neural networks (GNNs) that primarily aggregate information from local neighbors, PCMP prioritizes message propagation along higher-order path complexes. By incorporating longer paths, PCMP effectively captures long-range dependencies within the graph, enhancing its ability to model complex relationships.

A key feature of PCMP is the hierarchical message passing mechanism between different orders of path complexes. First, messages in higher-order paths are updated, reflecting the broader structure of the graph. These updated messages are then propagated to lower-order paths, ensuring that global information from longer paths informs to lower-order paths. After this, a feedback mechanism is employed, where updated messages from lower-order paths influence the higher-order paths, thus refining the representation at all levels. This bidirectional interaction between higher- and lower-order path allows PCMP to seamlessly integrate both global and local information effectively. The Path Message Passing Module can be found in Appendix A.2.

**Theorem 3.10.** *A Path Complex Message Passing (PCMP) with sufficient layers and injective neighborhood aggregators achieves the same expressive power as the PWL.*

## 4 EXPERIMENTS

### 4.1 BENCHMARK DATASETS AND MODELS

To thoroughly validate our PCMP model, we use three widely recognized benchmark datasets from MoleculeNet Wu et al. (2018) and MolBench Jiang et al. (2023). During data preprocessing, we employ the Merck molecular force field (MMFF94) function from RDKit to generate 3D molecular structures. The datasets are split into training, validation, and test sets using the scaffold splitting method, with a ratio of 8:1:1. Detailed descriptions of the datasets, preprocessing steps, and splitting method are provided in Appendix A.3.

We compare the performance of our PCMP model against state-of-the-art GNN models, both with and without pre-training. The non-pre-trained GNN models include (1) widely-used architectures such as GIN Xu et al. (2018), GAT Velickovic et al. (2017), and GCN Kipf & Welling (2016a); (2) recent models incorporating 3D molecular geometry, including SGCN Danel et al. (2020), DimeNet Gasteiger et al. (2020), and HMGNN Shui & Karypis (2020b); and (3) architectures specifically designed for molecular representation, such as D-MPNN Yang et al. (2019), AttentiveFP Xiong et al. (2019), and Mol-GDL Shen et al. (2023). For pre-trained models, we compare against N-Gram Liu et al. (2019), PretrainGNN Hu et al. (2019), GROVER Rong et al. (2020), GEM Fang et al. (2022), DMP Zhu et al. (2023), SMPT Li et al. (2024) and DGCL Jiang et al. (2024) .

Table 1: Comparison with GNN architectures. The best performance is indicated as **bold**, and the subindex indicates standard deviation values. * indicates that the result is not available for the model.

| | Method | QM7 | QM9 | Tox21 | HIV | MUV |
|---|---|---|---|---|---|---|
| GNN | GIN | $110.3_{(7.2)}$ | $0.00886_{(0.00005)}$ | $0.740_{(0.008)}$ | $0.753_{(0.019)}$ | $0.718_{(0.003)}$ |
| | GAT | $103.0_{(4.4)}$ | $0.01117_{(0.00018)}$ | $0.745_{(0.006)}$ | $0.724_{(0.008)}$ | $0.671_{(0.011)}$ |
| | GCN | $100.0_{(3.8)}$ | $0.00923_{(0.00019)}$ | $0.709_{(0.003)}$ | $0.740_{(0.003)}$ | $0.716_{(0.004)}$ |
| | D-MPNN | $103.5_{(8.6)}$ | $0.00812_{(0.00009)}$ | $0.759_{(0.007)}$ | $0.771_{(0.005)}$ | $0.786_{(0.014)}$ |
| | Attentive FP | $72.0_{(2.7)}$ | $0.00812_{(0.00001)}$ | $0.761_{(0.005)}$ | $0.757_{(0.014)}$ | $0.766_{(0.015)}$ |
| | GTransformer | $161.3_{(7.1)}$ | $0.00923_{(0.00019)}$ | * | * | * |
| | SGCN | $131.3_{(11.6)}$ | $0.01459_{(0.00055)}$ | * | * | * |
| | DimNet | $95.6_{(4.1)}$ | $0.01031_{(0.00076)}$ | * | * | * |
| | HMGNN | $101.6_{(3.2)}$ | $0.01239_{(0.00001)}$ | * | * | * |
| | Mol-GDL | $62.2_{(0.4)}$ | $0.00952_{(0.00013)}$ | $0.791_{(0.005)}$ | $0.808_{(0.007)}$ | $0.675_{(0.014)}$ |
| Pretrain_GNN | N-Gram$_{RF}$ | $92.8_{(4.0)}$ | $0.01037_{(0.00016)}$ | $0.743_{(0.004)}$ | $0.772_{(0.001)}$ | $0.769_{(0.007)}$ |
| | N-Gram$_{XGB}$ | $81.9_{(1.9)}$ | $0.00964_{(0.00031)}$ | $0.758_{(0.009)}$ | $0.787_{(0.004)}$ | $0.748_{(0.002)}$ |
| | PretrainGNN | $113.2_{(0.6)}$ | $0.00922_{(0.00004)}$ | $0.781_{(0.006)}$ | $0.799_{(0.007)}$ | $0.813_{(0.021)}$ |
| | GROVER$_{base}$ | $94.5_{(3.8)}$ | $0.00986_{(0.00055)}$ | $0.743_{(0.001)}$ | $0.625_{(0.009)}$ | $0.673_{(0.018)}$ |
| | GROVER$_{large}$ | $92.0_{(0.9)}$ | $0.00986_{(0.00025)}$ | $0.735_{(0.001)}$ | $0.682_{(0.011)}$ | $0.673_{(0.018)}$ |
| | MolCLR | $66.8_{(2.3)}$ | * | $0.750_{(0.002)}$ | $0.781_{(0.005)}$ | $0.796_{(0.019)}$ |
| | GEM | $58.9_{(0.8)}$ | $0.00746_{(0.00001)}$ | $0.781_{(0.001)}$ | $0.806_{(0.009)}$ | $0.817_{(0.005)}$ |
| | DMP | $74.4_{(1.2)}$ | * | $0.791_{(0.004)}$ | $0.814_{(0.004)}$ | * |
| | SMPT | * | * | $0.797_{(0.001)}$ | $0.812_{(0.001)}$ | $0.822_{(0.008)}$ |
| | DGCL | $100.9_{(2.5)}$ | * | $0.772_{(0.310)}$ | $0.815_{(1.100)}$ | * |
| | **PCMP** | $\mathbf{53.6}_{(2.1)}$ | $\mathbf{0.00683}_{(0.00005)}$ | $\mathbf{0.801}_{(0.002)}$ | $\mathbf{0.823}_{(0.004)}$ | $\mathbf{0.827}_{(0.015)}$ |

## 4.2 RESULTS

The comparison of our PCMP model with existing models on benchmark datasets is presented in Table 1. Detailed parameter settings for PCMP can be found in Section A.4 (Appendix A). Our PCMP model shows a significant performance advantage across datasets, mainly due to its advanced feature extraction capabilities and superior recognition of complex molecular structures. The message-passing mechanism in PCMP is organized into two distinct layers: the upper embedding, which considers upper adjacent neighbors, and the lower embedding, which incorporates both upper adjacent and face neighbors. This dual-layered approach integrates path information from multiple perspectives, enhancing the model's ability to capture both local and global graph structures. Each path is updated not only based on the features of its constituent nodes but also by incorporating information from both higher-order and lower-order connected paths. This sophisticated mechanism enables the model to detect subtle structural variations within molecules that are often difficult to distinguish. Compared to models that rely heavily on traditional pre-training, PCMP reduces computational demands by bypassing extensive pre-training phases while achieving excellent results, making it a promising solution for molecular property prediction.

## 4.3 ABLATION STUDY OF PCMP

**Impact of Message Passing Mechanisms** In the PCMP model, high-order features are first updated and then used as inputs to update low-order features, with the updated low-order features subsequently used to update high-order features. To evaluate the significance of this interaction, we introduced three variant models: PCMP-PARAL, PCMP-HL, and PCMP-LH, each limiting the

information exchange between different feature levels. Specifically, PCMP-PARAL restricts both high-to-low and low-to-high feature updates; PCMP-HL limits message passing from high-order to low-order features; and PCMP-LH restricts message passing from low-order to high-order features. Table 2 compares the performance of these models—PCMP-PARAL, PCMP-HL, PCMP-LH—against the standard PCMP on benchmark datasets.

Table 2: Results on benchmark datasets with different message passing mechanisms.

| Method | QM7 | QM9 | Tox21 | HIV | MUV |
|---|---|---|---|---|---|
| PCMP-PARAL | $56.9_{(1.5)}$ | $0.00751_{(0.00015)}$ | $0.779_{(0.008)}$ | $0.794_{(0.016)}$ | $0.808_{(0.006)}$ |
| PCMP-HL | $54.8_{(1.0)}$ | $0.00727_{(0.00006)}$ | $0.796_{(0.002)}$ | $0.803_{(0.020)}$ | $0.814_{(0.012)}$ |
| PCMP-LH | $55.3_{(1.9)}$ | $0.00764_{(0.00005)}$ | $0.793_{(0.004)}$ | $0.793_{(0.009)}$ | $0.806_{(0.004)}$ |
| PCMP | $\mathbf{53.6}_{(2.1)}$ | $\mathbf{0.00683}_{(0.00005)}$ | $\mathbf{0.801}_{(0.004)}$ | $\mathbf{0.823}_{(0.004)}$ | $\mathbf{0.827}_{(0.015)}$ |

**Impact of Input Path**   The PCMP model integrates various path-based features to improve its ability to accurately capture molecular structures. These paths, ranging from 0-path to 3-path, represent different levels of molecular interaction complexity, from the simplest to the most intricate. Table 3 shows the performance results for different path inputs on the benchmark datasets. As indicated, incorporating higher-order paths (2-path and 3-path) generally enhances the model's performance, with the inclusion of the 3-path yielding the best results across all datasets. This underscores its effectiveness in capturing complex molecular interactions. However, the slight performance degradation when excluding the 2-path and 3-path elements suggests that lower-order information remains crucial, especially for the QM7 dataset, where simpler molecular representations are sufficient.

Table 3: The results for input different path of the benchmark datasets.

| Input-Path | QM7 | QM9 | Tox21 | HIV | MUV |
|---|---|---|---|---|---|
| {0,1}-path | $57.0_{(1.4)}$ | $0.00898_{(0.00012)}$ | $0.786_{(0.006)}$ | $0.782_{(0.007)}$ | $0.767_{(0.003)}$ |
| {0,1,2}-path | $56.9_{(1.1)}$ | $0.00700_{(0.00006)}$ | $0.792_{(0.002)}$ | $0.803_{(0.005)}$ | $0.815_{(0.012)}$ |
| {0,1,2,3}-path | $\mathbf{53.6}_{(2.1)}$ | $\mathbf{0.00683}_{(0.00005)}$ | $\mathbf{0.801}_{(0.002)}$ | $\mathbf{0.823}_{(0.004)}$ | $\mathbf{0.827}_{(0.015)}$ |

**Impact of Readout Path**   To investigate whether feature outputs at different levels can improve model performance, we designed several output strategies. As shown in Table 4, utilizing outputs from multiple levels allows the PCMP model to capture both low-order and high-order molecular features, significantly enhancing the model's ability to represent complex structures and improving its overall expressiveness.

Table 4: The results for Readout different path of the benchmark datasets.

| Output-Path | QM7 | QM9 | Tox21 | HIV | MUV |
|---|---|---|---|---|---|
| {0}-path | $56.8_{(1.2)}$ | $0.00797_{(0.00004)}$ | $0.789_{(0.008)}$ | $0.784_{(0.010)}$ | $0.806_{(0.018)}$ |
| {0,1}-path | $56.4_{(1.5)}$ | $0.00698_{(0.00010)}$ | $0.793_{(0.009)}$ | $0.800_{(0.007)}$ | $0.808_{(0.007)}$ |
| {0,1,2}-path | $\mathbf{53.6}_{(2.1)}$ | $\mathbf{0.00683}_{(0.00005)}$ | $\mathbf{0.801}_{(0.002)}$ | $\mathbf{0.823}_{(0.004)}$ | $\mathbf{0.827}_{(0.015)}$ |

**Visualization of the High-order Path Contribute**   To clearly quantify the impact of different high-order paths in the PCMP model on both message passing and Readout processes, we employ various order paths in regression tasks (QM7 and QM9 datasets) and classification tasks (Tox21, HIV, and MUV datasets). Figure 4 displays the performance improvement. It is important to note that the PCMP model utilizes face and coffee neighbors for information dissemination and update processes. Therefore, the paths we have selected to analyze are the 0,1-order path, 0,1,2-order path, and 0,1,2,3-order path. This delineation helps in understanding how path complexity influences the effectiveness of the model across different types of tasks and datasets.

**Impact of Geometric Feature in Path Complex**   Since the path complex uses different geometric features for different order paths in addition to the band angle and dihedral angle in MD in characterizing the characteristics of molecules, the effectiveness of these added geometric features is verified

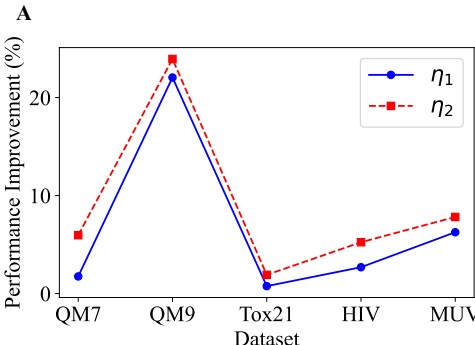 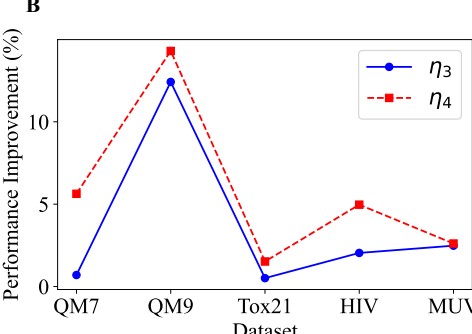

Figure 4: Visualization of High-order Path Contributions Across Datasets. A: Input high-order paths showing model performance improvement, where $\eta_1$ and $\eta_2$ represent the improvements with input 0,1,2-order paths and 0,1,2,3-order 0,1,2-order paths relative to input 0,1-order paths. B: Output high-order paths and model performance improvement, where $\eta_3$ and $\eta_4$ represent the improvements with output paths 0,1-order paths and 0,1,2-order paths relative to output 0-order path.

in Table 5. By comparison, we found that after using these geometric features, QM7 increased by 4.97%, QM9 increased by 5.92%, Tox21 increased by 3.49%, HIV increased by 2.88%, and MUV increased by 6.71%.

Table 5: Comparison of features. Angle and Dihedral represent that 2-path and 3-path only use angle and Dihedral, and Path feature represents the encoding method in Table7.

| | Method | QM7 | QM9 | Tox21 | HIV | MUV |
|---|---|---|---|---|---|---|
| PCMP | Angle and Dihedral | $56.4_{(1.7)}$ | $0.00726_{(0.00012)}$ | $0.774_{(0.009)}$ | $0.800_{(0.006)}$ | $0.775_{(0.023)}$ |
| | Path Features | $\mathbf{53.6}_{(2.1)}$ | $\mathbf{0.00683}_{(0.00005)}$ | $\mathbf{0.801}_{(0.002)}$ | $\mathbf{0.823}_{(0.004)}$ | $\mathbf{0.827}_{(0.015)}$ |

**Sensitivity of Hyperparameters** We explored the model's sensitivity to hyperparameters and the experimental results are displayed in Table 6. According to the results, the model's performance metrics are generally stable across different hyperparameters settings.

Table 6: Sensitivity of hyperparameters for benchmark datasets.

| Hyperparameters | | QM7 | QM9 | Tox21 | HIV | MUV |
|---|---|---|---|---|---|---|
| Head | 1 | $\mathbf{53.6}_{(2.1)}$ | $0.00931_{(0.00007)}$ | $0.748_{(0.006)}$ | $0.793_{(0.004)}$ | $\mathbf{0.827}_{(0.015)}$ |
| | 2 | $54.7_{(1.8)}$ | $0.00747_{(0.00006)}$ | $0.754_{(0.007)}$ | $\mathbf{0.823}_{(0.004)}$ | $0.787_{(0.019)}$ |
| | 4 | $55.9_{(1.0)}$ | $0.00721_{(0.00006)}$ | $0.759_{(0.005)}$ | $0.799_{(0.013)}$ | $0.801_{(0.017)}$ |
| | 6 | $58.3_{(2.5)}$ | $\mathbf{0.00683}_{(0.00005)}$ | $\mathbf{0.801}_{(0.004)}$ | $0.807_{(0.016)}$ | $0.814_{(0.013)}$ |
| | 8 | $58.5_{(1.2)}$ | $0.00826_{(0.00008)}$ | $0.765_{(0.004)}$ | $0.808_{(0.004)}$ | $0.783_{(0.008)}$ |
| Batch Size | 64 | $59.6_{(1.7)}$ | $0.00868_{(0.00012)}$ | $0.784_{(0.004)}$ | $0.796_{(0.003)}$ | $0.785_{(0.007)}$ |
| | 128 | $57.5_{(1.0)}$ | $0.01040_{(0.00007)}$ | $0.778_{(0.003)}$ | $0.803_{(0.007)}$ | $0.819_{(0.016)}$ |
| | 256 | $54.3_{(0.9)}$ | $0.00756_{(0.00014)}$ | $0.778_{(0.003)}$ | $0.807_{(0.015)}$ | $0.789_{(0.022)}$ |
| | 512 | $\mathbf{53.6}_{(2.1)}$ | $\mathbf{0.00683}_{(0.00004)}$ | $\mathbf{0.801}_{(0.004)}$ | $\mathbf{0.823}_{(0.004)}$ | $\mathbf{0.827}_{(0.015)}$ |
| LR | 5e-3 | $58.5_{(1.2)}$ | $0.00784_{(0.00004)}$ | $0.782_{(0.004)}$ | $0.805_{(0.009)}$ | $0.808_{(0.014)}$ |
| | 1e-3 | $56.2_{(1.4)}$ | $\mathbf{0.00683}_{(0.00005)}$ | $\mathbf{0.791}_{(0.009)}$ | $0.811_{(0.013)}$ | $\mathbf{0.827}_{(0.015)}$ |
| | 5e-4 | $54.7_{(1.3)}$ | $0.00880_{(0.00006)}$ | $0.781_{(0.012)}$ | $\mathbf{0.823}_{(0.005)}$ | $0.814_{(0.012)}$ |
| | 1e-4 | $\mathbf{53.6}_{(2.1)}$ | $0.00962_{(0.00012)}$ | $0.789_{(0.006)}$ | $0.799_{(0.018)}$ | $0.822_{(0.013)}$ |
| | 5e-5 | $64.9_{(3.1)}$ | $0.00784_{(0.00011)}$ | $0.724_{(0.007)}$ | $0.788_{(0.024)}$ | $0.818_{(0.019)}$ |

## 5 CONCLUSION

In this study, we introduced the path complex message passing, a novel model for molecular structure representation based on path complexes, designed to predict molecular properties. By integrating force fields with path complexes, the model enhances our understanding of the relationship between molecular structure and function, offering valuable insights for both theoretical research and practical applications in molecular design and materials science. The PCMP model employs 0-paths for atomic properties, 1-paths for pairwise interactions, 2-paths for bond angle terms, and 3-paths for dihedral angle information. These paths are used to compute attention scores, enabling efficient message propagation and feature integration across various levels of molecular information. Validation on five benchmark datasets has demonstrated the PCMP's superior predictive capabilities. Ablation studies further confirm that incorporating higher-order features significantly improves performance, pointing to promising directions for future research in molecular simulation and design.

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

## A  APPENDIX / SUPPLEMENTAL MATERIAL

### A.1  INITIALIZATION FEATURES

Table 7: MD Encoder for Path Features

| Features Type | | Description | Type | Size |
|---|---|---|---|---|
| 1-Path (bond) | Bond Directionality | None, Beginwedge, Begindash, etc. | One-Hot | 7 |
| | Bond Type | Single, Double, Triple, or Aromatic. | One-Hot | 4 |
| | Bond Length | Numerical length of the bond. | Float | 1 |
| | In Ring | Indicates if the bond is part of a chemical ring. | One-Hot | 2 |
| 1-Path (non-bond) cutoff=3 | Atom charges | Atoms charges in Molecular $(q_i, q_j, q_i \cdot q_j)$ | Float | 3 |
| | Distance between atoms | Distance between atoms $(1/d_{ij}, 1/d_{ij}^6, 1/d_{ij}^{12})$ | Float | 3 |
| 2-Path | Centroid distance | Centroid position of the triangle formed by 2-path | Float | 3 |
| | Distance | Three bond lengths (two for covalent bond and one for non-covalent bond) | Float | 3 |
| | Area | Triangle area spanned by 2-path | Float | 1 |
| | Bond Angle | Bond angle for 2-path | Float | 1 |
| 3-Path | Volume | Volume spanned by 3-path | Float | 1 |
| | Dihedral | Dihedral angle for 3-path | Float | 1 |
| | Total Area | Total Area of the corresponding four triangles | Float | 1 |
| | Bond Length | Non-covalent bond length $(\{v_1v_3\},\{v_2v_4\}, \{v_1v_4\})$ | Float | 3 |

As depicted in Figure 5, the $C_2H_6O$ molecule is illustrated alongside its corresponding path simplices. Specifically, our 1-path simplex captures bond lengths, the 2-path simplex details bond angles, and the 3-path simplex reflects dihedral angles.

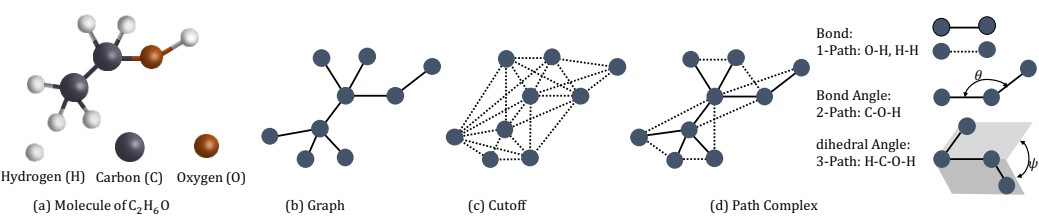

Hydrogen (H)  Carbon (C)  Oxygen (O)

(a) Molecule of $C_2H_6O$   (b) Graph   (c) Cutoff   (d) Path Complex

Figure 5: Different Representations of the $C_2H_6O$ Molecule. (a) displays the molecular structure of $C_2H_6O$, including the oxygen (O), carbon (C), and hydrogen (H) atoms. (b) shows the graph representation based on chemical bonds. (c) illustrates the nearly fully connected graph generated based on a distance threshold (cutoff). (d) presents the representation using the path complex method and its physical implications. In the diagrams, solid lines represent chemical bonds, while dashed lines represent cutoff connections.

To fully incorporate MD force field information into molecular representation, we propose molecular path complex, which uses path simplices at different dimensions to explicitly characterize force field (covalent) bond terms. More specifically, our 1-path simplex represents bond length information, 2-path simplex describe bond angles, and 3-path simplex characterizes dihedral angle.

## A.2 Path message-passing module

A central component of our PCMP model is path (simplex) Grigor'yan et al. (2024) message-passing module, where path features are updates based on path neighbors (same order paths), cofaces (higher-order paths), and faces (lower-order paths). Mathematically, each $n$-path will always have two unique $(n-1)$-faces, but many $n$-path neighbors and $(n+1)$-cofaces. In our PCMP framework, the simplex message-passing module contains two parts, i.e., message embedding and message updating. Two message embedding modules, i.e., upper embedding and lower embedding, are considered. In upper embedding module, path message will be generated using its neighbors and cofaces, while for lower embedding module, path messages will be generated using its neighbors and faces. The path feature will be updated using path messages from both upper and lower embedding through a message updating module. An illustration of our PCMP module is shown in Figure 6.

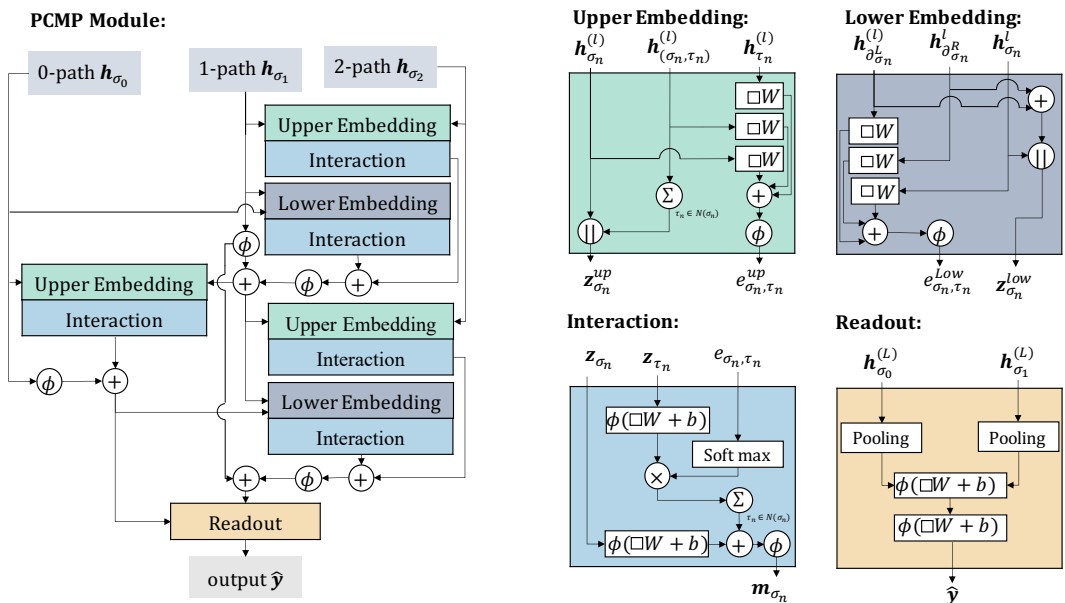

Figure 6: The PCMP Module. $\square$ denotes the layer's input, $\|$ concatenation, and $\phi$ a non-linearity. Upper embedding and Upper interaction refer to utilizing high-order path features to update low-order path features, while Lower embedding and Lower interaction refer to using low-order path features to update high-order path features.

The upper embedding module generates path message based on path neighbors and cofaces. For an $n$-path $\sigma_n$ and its neighbors $\tau_n$, we denote their path feature vectors as $\boldsymbol{h}_{\sigma_n}$ and $\boldsymbol{h}_{\tau_n}$ respectively. The common coface of $\sigma_n$ and $\tau_n$ is denoted as $(\tau_n, \sigma_n)$ and $\boldsymbol{h}_{(\tau_n, \sigma_n)}$ the associate path feature. The upper attention score $e^{up}_{\sigma_n, \tau_n}$ and upper concatenated feature $\boldsymbol{z}^{up}_{\sigma_n}$ can be expressed as,

$$\boldsymbol{f}_{\sigma_n} = \boldsymbol{h}_{\sigma_n} \mathbf{W}^{up}_n, \quad \boldsymbol{f}_{\tau_n} = \boldsymbol{h}_{\tau_n} \mathbf{W}^{up}_n, \quad \boldsymbol{f}_{(\sigma_n, \tau_n)} = \boldsymbol{h}_{(\sigma_n, \tau_n)} \mathbf{W}_{n+1},$$

$$e^{up}_{\sigma_n, \tau_n} = \text{ReLU}(\boldsymbol{f}_{\sigma_n} + \boldsymbol{f}_{\tau_n} + \boldsymbol{f}_{(\sigma_n, \tau_n)}), \quad \boldsymbol{z}^{up}_{\sigma_n} = [\, \boldsymbol{h}_{\sigma_n} \, \| \sum_{\tau_n \in \mathcal{N}(\sigma_n)} \frac{1}{|\mathcal{N}(\sigma_n)|} \boldsymbol{h}_{(\sigma_n, \tau_n)} \,],$$

where ReLU is a non-linear activation function and $\mathbf{W}^{up}_n$ and $\mathbf{W}_{n+1}$ are weight matrices. Note that $\|$ is the concatenation operator, $\mathcal{N}(\sigma_n)$ denotes the neighbors of path $\sigma_n$, and $|\mathcal{N}(\sigma_n)|$ is the total number of neighbors of path $\sigma_n$.

The lower embedding module generates path message based on path neighbors and faces. For an $n$-path $\sigma_n$, we use $\partial^R_{\sigma_n}$ and $\partial^L_{\sigma_n}$ to represent its right and left faces. The lower attention score $e^{low}_{\sigma_n, \tau_n}$

and lower concatenated feature $z_{\sigma_n}^{low}$ can be expressed as,

$$\boldsymbol{f}_{\sigma_n} = \boldsymbol{h}_{\sigma_n}\mathbf{W}_n^{low}, \quad \boldsymbol{f}_{\partial_{\sigma_n}^L} = \boldsymbol{h}_{\partial_{\sigma_n}^L}\mathbf{W}_{n-1}, \quad \boldsymbol{f}_{\partial_{\sigma_n}^R} = \boldsymbol{h}_{\partial_{\sigma_n}^R}\mathbf{W}_{n-1},$$

$$e_{\sigma_n,\tau_n}^{low} = \text{ReLU}(\boldsymbol{f}_{\partial_{\sigma_n}^L} + \boldsymbol{f}_{\partial_{\sigma_n}^R} + \boldsymbol{f}_{\sigma_n}), \quad z_{\sigma_n}^{low} = [\,(\boldsymbol{h}_{\partial_{\sigma_n}^L} + \boldsymbol{h}_{\partial_{\sigma_n}^R}) \parallel \boldsymbol{h}_{\sigma_n}\,],$$

where $\mathbf{W}_n^{low}$ and $\mathbf{W}_{n-1}$ are weight matrices.

The path feature is updated by using message from both upper embedding and low embedding. First, upper and lower path message is generated from the upper embedding and low embedding respectively as follows,

$$\boldsymbol{c}_{\sigma_n}^{up/low} = \text{ReLU}(z_{\sigma_n}^{up/low}\mathbf{W}_n^{up/low} + b^{up/low}), \quad \alpha_{\sigma_n,\tau_n}^{up/low} = \frac{e_{\sigma_n,\tau_n}^{up/low}}{\sum_{\kappa_n \in \mathcal{N}(\sigma_n)} e_{\sigma_n,\kappa_n}^{up/low}},$$

$$\boldsymbol{m}_{\sigma_n}^{up/low} = \text{LeakyReLU}(\boldsymbol{c}_{\sigma_n}^{up/low} + \sum_{\tau_n \in \mathcal{N}(\sigma_n)} \alpha_{\sigma_n,\tau_n}^{up/low}\boldsymbol{c}_{\tau_n}^{up/low}),$$

then path feature is updated by using both messages as follows,

$$\boldsymbol{h}_{\sigma_n}^{(l+1)} = \text{LeakyReLU}((\boldsymbol{m}_{\sigma_n}^{low})^{(l)} + (\boldsymbol{m}_{\sigma_n}^{up})^{(l)}).$$

Note that $h_{\sigma_n}^{(l+1)}$ means the updated feature vector for $n$-path $\sigma_n$ at the $(l+1)$-th layer. It depends on the upper and lower message information at the $l$-th layer.

### A.3 DATASET DETAILS, MIN-MAX SCALING, SPLITTING METHO AND MEAN ABSOLUTE ERR

In this study, we analyzed five datasets from MoleculeNet Wu et al. (2018) and MolBench Jiang et al. (2023): QM7 Blum & Reymond (2009), QM9 Ruddigkeit et al. (2012), Tox21, Hiv and Muv, all of which are publicly available on the MoleculeNet website: https://moleculenet.org/datasets-1. Details about these datasets are in Table 8. Note that the subindex indicates standard deviation val-

Table 8: The details of the datasets. Note that the subindex indicates standard deviation values.

| Dataset | QM7 | QM9 | Tox21 | HIV | MUV |
|---|---|---|---|---|---|
| No. molecules | 6,830 | 133,885 | 7831 | 41127 | 93808 |
| No. average atoms | $16_{(3)}$ | $18_{(3)}$ | $36_{(23)}$ | $46_{(24)}$ | $43_{(10)}$ |
| No. tasks | 1 | 3 | 12 | 1 | 17 |
| Task type | Regression | Regression | Classification | Classification | Classification |
| Evaluation | MAE | MAE | ROC-AUC | ROC-AUC | ROC-AUC |

ues. For instance, the element $16_{(13)}$ means the number of average atoms in QM7 is 16, with 13 as its standard deviation. The QM7 dataset is a subset of the GDB-13 database Blum & Reymond (2009), which contains approximately 1 billion organic molecules with up to seven "heavy" atoms (C, N, O, S). The QM9 dataset, a subset of the GDB-17 database, provides twelve properties, encompassing geometric, energetic, electronic, and thermodynamic properties, following the baseline methods PCMP use the electronic spatial extent ($\alpha$), and the energies of the highest occupied molecular orbital ($\epsilon$HOMO) and the lowest unoccupied molecular orbital ($\epsilon$LUMO) as the targets. Tox21 is qualitative toxicity measurements on 12 biological targets, including nuclear receptors and stress response pathways. HIV is experimentally measured abilities to inhibit HIV replication. MUV is subset of PubChem BioAssay by applying a refined nearest neighbor analysis, designed for validation of virtual screening techniques.

**Min-Max Scaling**   Given that QM7 and QM9 involve regression, we applied min-max normalization to scale target values between 0 and 1. In multiple-target regression tasks, Min-Max Scaling is commonly used to normalize the targets. This technique linearly transforms the target values to a specified range between a minimum and maximum value. The transformation follows the formula:

$$\bar{y} = \frac{y - y_{\min}}{y_{\max} - y_{\min}}, \quad y_{\text{scal}} = y_{\max} - y_{\min} \tag{1}$$

Here, $\overline{y}$ represents the normalized target value, $y$ is the original target value, $y_{\min}$ is the minimum value of the target, and $y_{\max}$ is the maximum value of the target.

During prediction, the normalized predictions obtained from the model need to be transformed back to the original scale of the target values. The transformation is performed using the formula:

$$\tilde{y} = \hat{y} \cdot y_{\text{scal}} + y_{\min}, \quad y = \overline{y} \cdot y_{\text{scal}} + y_{\min} \tag{2}$$

where $\hat{y}$ is the model output, and $\tilde{y}$ and $y$ are used for loss function computation and evaluation.

This normalization process ensures that all target values are scaled within a fixed range, typically between 0 and 1. It facilitates better convergence during model training and helps in handling targets with varying scales effectively. Furthermore, Min-Max Scaling maintains the relative relationships between target values while bringing them into the desired range, making it a suitable choice for multiple-target regression tasks.

**Splitting Method**   Following the work of Bharath Ramsundar Ramsundar et al. (2019), we employed scaffold splitting to partition all datasets. This method segments molecules based on their scaffolds (molecular substructures). Scaffold splitting is a more challenging partitioning approach that can better evaluate a model's generalization ability on out-of-distribution data samples. To ensure a fair comparison with other models, we adopted the same scaffold splitting method to divide the task datasets into training, validation, and test sets with a ratio of 8:1:1.

**MAE (Mean Absolute Error)**   The Mean Absolute Error (MAE) is defined as:

$$MAE = \frac{1}{N} \sum_{i=1}^{N} |y_i - \hat{y}_i| \tag{3}$$

where $y_i$ and $\tilde{y}_i$ represent the true value and predicted value of the $i^{th}$ sample respectively. MAE is a commonly used metric for evaluating regression performance. A lower MAE value indicates higher prediction accuracy, with a decrease in MAE typically suggesting improved model performance.

Table 9: Hyperparameters set up.

| Dataset | QM7 | QM9 | Tox21 | HIV | MUV |
|---|---|---|---|---|---|
| Learning rate | 1e-4 | 1e-3 | 1.5e-4 | 1e-3 | 1e-4 |
| Batch size | 512 | 512 | 512 | 512 | 512 |
| No.heads | 1 | 6 | 6 | 2 | 1 |
| No.layers | 2 | 2 | 2 | 2 | 2 |
| Train/Valid/Test | 8:1:1 | 8:1:1 | 8:1:1 | 8:1:1 | 8:1:1 |
| Loss function | L1 | L1 | BCE | BCE | BCE |
| Optimizer | ADAM | ADAM | ADAM | ADAM | ADAM |
| Epochs | 500 | 500 | 1000 | 1000 | 1000 |
| Seed | 42 | 42 | 42 | 42 | 42 |

## A.4 HYPERPARAMETERS SETUP

We have set up a set of hyperparameters for training the model are summarized in Table 9. Inaddition, the optimizer selected as ADAM, and the loss function chosen as L1. All models are trained using NVIDIA RTX A5000 32GB GPUs. The running times are show in the table

## B MATHEMATICAL ANALYSIS OF PATH COMPLEX

### B.1 PATH COMPLEX

**Definition B.1** (Elementary path Grigor'yan et al. (2012)). Given a set $V$, an elementary $n$-path of $V$ is any sequence of $n + 1$ elements $v_0 v_1 \cdots v_n$ of $V$, denoted by $\sigma_n = v_0 v_1 \cdots v_n$

**Definition B.2** (Path complex Grigor'yan et al. (2012)). A path complex $P$ over the vertex set $V$ is a collection of elementary paths of $V$ such that $\forall \sigma_n = v_0 v_1 \cdots v_n \in P$, $v_1 \cdots v_n \in P$, $v_0 v_1 \cdots v_{n-1} \in P$.

The element $\sigma_n$ of $P$ that has $n+1$ vertices is called an $n$-path of $P$. The path $\sigma$ is called a face of the path $\tau$ if $\sigma$ is derived from $\tau$ by removing the first or last vertex. The $n$-path $\tau_n$ is called a coface of $(n-1)$-path $\sigma_{n-1}$ if $\sigma_{n-1}$ is a face of $\tau_n$. Two $n$-paths are upper adjacent if they are faces of a common $(n+1)$-path, lower adjacent if they have a common $(n-1)$-path as face. For an $n$-path $\sigma_n$, let $\mathcal{B}(\sigma_n)$ be the set of faces of $\sigma_n$, $\mathcal{C}(\sigma_n)$ be the set of cofaces of $\sigma_n$, $\mathcal{N}_\uparrow(\sigma_n)$ be the set of $n$-paths that are upper adjacent with $\sigma_n$, $\mathcal{N}_\downarrow(\sigma_n)$ be the set of $n$-paths that are lower adjacent with $\sigma_n$. Note that we can use the above four relations, including face-relation, coface-relation, upper adjacency and lower adjacency, to define the neighbors of an $n$-path $\sigma_n$. We give construction of path complex from graphs, simplicial complexes and hypergraphs. We give construction of path complex from graphs, simplicial complexes and hypergraphs.

### B.1.1 PATH COMPLEX FROM GRAPHS

**Definition B.3** (Path). Given an undirected graph $G = (V, E)$ over the verset set $V$, we define the $n$-path $\sigma_n$ of $G$ as any sequence of $n+1$ vertices $v_0 v_1 \cdots v_n (v_i \in V)$ satisfying the following conditions:

1. $\forall i (0 \leqslant i < n), (v_i, v_{i+1}) \in E$ or $(v_{i+1}, v_i) \in E$.

2. $\forall i \neq j, v_i \neq v_j$.

Note that for each $n$-path $\sigma_n = v_0 v_1 \cdots v_n$, $\sigma_n' = v_n \cdots v_1 v_0$ is also an $n$-path, we identify these two paths as the same one.

**Definition B.4** (Path complex from graphs). Given an undirected graph $G = (V, E)$, let $P_n$ be the set of all $n$-paths of $G$, then $P_G = \bigcup_n P_n$ form a path complex. We call $P_G$ the path complex derived from $G$.

It can be seen that the path complex $P_G$ derived from $G$ is determined by $P_0$ and $P_1$.

**Theorem B.5** (Path Complex Invariance). *The graph neural network is invariant to the permutation of the simplexes in the path complex $P_G$, meaning the output of the network remains unchanged under any permutation $\pi$ of the vertices.*

### B.1.2 PATH COMPLEX FROM SIMPLICIAL COMPLEX

**Definition B.6** (Simplicial complex). A simplicial complex $\mathcal{K}$ over the vertex set $V$ is a collection of vertex subsets of $V$ satisfying that if $\sigma \in \mathcal{K}$, $\tau \subset \sigma$, $\tau \in \mathcal{K}$.

The element $\sigma_k$ of $\mathcal{K}$ that has $k+1$ vertices is called an $k$-simplex. A simplex $\sigma$ is called a face of the simplex $\tau$ and $\tau$ is called a coface of $\sigma$ if $\sigma \subset \tau$. For any two $k$-simplices $\sigma_k, \tau_k \in \mathcal{K}$, $\sigma_k$ and $\tau_k$ are called upper adjacent if they are both faces of an $(k+1)$-simplex $\alpha_{k+1} \in \mathcal{K}$. Two $k$-simplices $\sigma_k, \tau_k$ are called lower adjacent if they share a common $(k-1)$-simplex as faces.

**Definition B.7** (Path). Given a simplicial complex $\mathcal{K}$, we define an $(k, n)$-path $e_n^k$ of $\mathcal{K}$ as a sequence of $n+1$ $k$-simplices $\sigma_k^0 \sigma_k^1 \cdots \sigma_k^n$ satisfying the following conditions:

1. $\forall i (0 \leqslant i < n), \sigma_k^i$ and $\sigma_k^{i+1}$ are upper adjacent.

2. $\forall i \neq j, \sigma_k^i \neq \sigma_k^j$

We can also use lower adjacent, face and the coface relation to define paths. Note that for each $(k, n)$-path $\sigma_k^0 \sigma_k^1 \cdots \sigma_k^n$, there is an $(k, n)$-path $\sigma_k^n \cdots \sigma_k^0$. We identify these two paths as the same one.

**Definition B.8** (Path complex from simplicial complex). Given a simplicial complex $\mathcal{K}$, let $P_n^k$ be the set of all $(k, n)$-paths of $\mathcal{K}$, then $P_\mathcal{K}^k = \bigcup_n P_n^k$ form a path complex.

For the simplicial complex $\mathcal{K}$, its one-skeleton forms a graph $\mathcal{K}_1$, we have $P_{\mathcal{K}_1} = P_\mathcal{K}^0$.

### B.1.3 Path complex from hypergraphs

**Definition B.9** (Hypergraph). A hypergraph $\mathcal{H}$ over the vertex set $V$ is a collection of vertex subsets of $V$.

The element $\sigma_k$ of $\mathcal{H}$ that has $k + 1$ vertices is called an $k$-hyperedge. Two hyperedges are called lower adjacent if their intersection is not empty.

**Definition B.10** (Path). Given a hypergraph $\mathcal{H}$ over the vertex set $V$, we define an $n$-path of $\mathcal{H}$ as a sequence of $n + 1$ hyperedges $\sigma^0 \sigma^1 \cdots \sigma^n$ such that any two adjacent hyperedges are lower adjacent and any two hyperedges are not same.

**Definition B.11** (Path complex from hypergraphs). Given a hypergraph $\mathcal{H}$, let $P_n$ be the set of all $n$-paths of $\mathcal{H}$, then $P_\mathcal{H} = \bigcup_n P_n$ form a path complex.

### B.2 Homology of Path Complex

The homology of path complex is a new homology theory that breaks the landscape of classical homology theory in algebraic topology, introducing a new framework for exploring the topology of more general mathematical structures Grigor'yan et al. (2012); Grigor'yan et al. (2014; 2020). This homology theory was initially called path homology and renamed GLMY homology in 2022, which advances the study of topological foundations for complex networks Chowdhury & Mémoli (2018); Chowdhury et al. (2022) and has been successfully applied in complex disease Wu et al. (2023), biology and material sciences Chen et al. (2023). Next, we give the construction of homology of path complexes.

Given a path complex $P$ over $V$, We fix a field coefficient $\mathbb{F}$, let $\Lambda_n(P)$ be the vector space spanned by all the elementary $n$-paths of $P$. Considering the standard boundary operator $\partial_n : \Lambda_n(P) \to \Lambda_{n-1}(P)$

$$\forall \sigma_n = v_0 v_1 \cdots v_n \in P, \qquad \partial_n(\sigma_n) = \sum_{i=0}^{n} (-1)^i v_0 \cdots v_{i-1} v_{i+1} \cdots v_n$$

We have $\partial_n \partial_{n+1} = 0$. Let $\mathcal{A}_n(P)$ be the vector space spanned by all the $n$-paths of $P$, usually $\partial(\mathcal{A}_n) \not\subset \mathcal{A}_{n-1}(P)$. We consider the following subspace $\Omega_n(P)$ of $\mathcal{A}_n(P)$

$$\Omega_n(P) = \{u \in \mathcal{A}_n(P) | \partial(u) \in \mathcal{A}_{n-1}(P)\}$$

Then we have $\partial_n(\Omega_n(P)) \subset \Omega_{n-1}(P)$. Consequently, we get a chain complex $(\Omega_*(P), \partial_*)$

$$\cdots \to \Omega_{n+1}(P) \xrightarrow{\partial_{n+1}} \Omega_n(P) \xrightarrow{\partial_n} \Omega_{n-1}(P) \to \cdots$$

**Definition B.12** (Homology of path complex). Given a path complex $P$, its $k$-homology is defined as the $k$-th homology of the chain complex $(\Omega_*(P), \partial_*)$

$$\mathrm{H}_k(P) = \mathrm{H}_k((\Omega_*, \partial_*))$$

This definition can be directly applied to the path complexes derived from graphs, simplicial complexes and hypergraphs.

### B.3 Weak Isomorphism Invariance of the Path Complex Homology

For an undirected graph $G = (V, E)$, the degree of a vertex $v \in V$ is the number of edges that contain $v$ and we denoted it by $deg(v)$.

**Definition B.13** (Graph collapse and expansion). Given a graph $G = (V, E)$, take an edge $(v_1, v_2) \in E$ such that $deg(v_1) = 1$. Let $V' = V \backslash \{v_1\}$, $E' = E \backslash \{(v_1, v_2)\}$, then $G' = (V', E')$ is a new graph. We say that $G'$ is derived from $G$ by a graph collapse and $G$ is derived from $G'$ by a graph expansion.

**Definition B.14** (Weak isomorphic). Given two graphs $G_1, G_2$, $G_1$ and $G_2$ are called weak isomorphic if $G_1$ can be derive from $G_2$ by a sequence of graph collapse and expansion operations.

It is obvious that two graphs are weak isomorphic if they are isomorphic.

**Theorem B.15.** *If two graphs $G_1$ and $G_2$ are weak isomorphic, then*

    *1. The number of connected components of $G_1$ and $G_2$ are same.*

    *2. The number of cycles of $G_1$ and $G_2$ are same.*

*Proof.* Let $G_1 = (V_1, E_1), G_2 = (V_2, E_2)$, without loss of generality, we can assume that $G_2$ is derived by collapsing an edge $(v_1, v_2) \in E_1$ from $G_1$ and $deg(v_1) = 1$.

    1. This is obvious.

    2. Let $C(G_i)$ be the set of cycles of $G_i$, then we have $C(G_2) \subset C(G_1)$ because $G_2$ is a subgraph of $G_1$. $\forall c \in C(G_1)$, $c$ is a sequence of vertices such that the degree of each vertex is 2. So $v_1$ is not contained in $c$, $c \in C(G_2)$. Consequently,

$$C(G_1) = C(G_2)$$

$\square$

**Theorem B.16.** *Given two graphs $G_1, G_2$, let $P_{G_1}, P_{G_2}$ be the path complexes derived from $G_1$ and $G_2$ respectively. If $G_1$ and $G_2$ are weak isomorphic, then*

$$\mathrm{H}_k(P_{G_1}) \cong \mathrm{H}_k(P_{G_2})\ (k \geqslant 0)$$

.

*Proof.* Let $G_1 = (V_1, E_1), G_2 = (V_2, E_2)$, without loss of generality, we can assume that $G_2$ is derived by collapsing an edge $(v_1, v_2) \in E_1$ from $G_1$ and $deg(v_1) = 1$.

    1. $k = 0$
$$\Omega_0(P_{G_1}) = <v_1> \oplus \Omega_0(P_{G_2}), \quad \Omega_0(P_{G_2}) = <v|\ v \in V_2>$$
$$\Omega_1(P_{G_1}) = <v_1v_2> \oplus \Omega_1(P_{G_2}), \quad \Omega_1(P_{G_2}) = <e|\ e \in E_2>$$

We have $Ker\partial|_{\Omega_0(P_{G_1})} = <v_1> \oplus Ker\partial|_{\Omega_0(P_{G_2})}$, $Im\partial|_{\Omega_1(P_{G_1})} = <v_2 - v_1> \oplus Im\partial|_{\Omega_1(P_{G_2})} = <v_1> \oplus Im\partial|_{\Omega_1(P_{G_2})}$. Consequently,

$$\begin{aligned}\mathrm{H}_0(P_{G_1}) &= \frac{Ker\partial|_{\Omega_0(P_{G_1})}}{Im\partial|_{\Omega_1(P_{G_1})}} \\ &= \frac{<v_1> \oplus Ker\partial|_{\Omega_0(P_{G_2})}}{<v_1> \oplus Im\partial|_{\Omega_1(P_{G_2})}} \\ &= \frac{Ker\partial|_{\Omega_0(P_{G_2})}}{Im\partial|_{\Omega_1(P_{G_2})}} \\ &= \mathrm{H}_0(P_{G_2})\end{aligned}$$

    2. $k = 1$
$$\Omega_1(P_{G_1}) = <v_1v_2> \oplus \Omega_1(P_{G_2}), \quad \Omega_1(P_{G_2}) = <e|\ e \in E_2>$$

The degree of vertex $v_1$ is one means $v_1$ only appears in the 1-path $(v_1, v_2)$, so $(v_1, v_2)$ cannot be contained in the kernel of $\partial$ on $\Omega_1(P_{G_1})$, which means that

$$Ker\partial|_{\Omega_1(P_{G_1})} = Ker\partial|_{\Omega_1(P_{G_2})}$$

$$\mathcal{A}_2(P_{G_1}) = <v_1v_2v|v_2 \neq v \in V_2> \oplus \mathcal{A}_2(P_{G_2})$$

Note that $(v_1, v)$ is not a 1-path for any $v \in V_2(v \neq v_2)$, so

$$\Omega_2(P_{G_1}) = \Omega_2(P_{G_2})$$

Consequently,

$$\mathrm{H}_1(P_{G_1}) = \frac{Ker\partial|_{\Omega_1(P_{G_1})}}{Im\partial|_{\Omega_2(P_{G_1})}}$$

$$= \frac{Ker\partial|_{\Omega_1(P_{G_2})}}{Im\partial|_{\Omega_2(P_{G_2})}}$$

$$= \mathrm{H}_1(P_{G_2})$$

3. $k \geqslant 2$. It suffices to prove that

$$\Omega_k(P_{G_1}) = \Omega_k(P_{G_2}) \, (k \geqslant 2)$$

It is obvious that $\Omega_k(P_{G_2}) \subset \Omega_k(P_{G_1})$. So we only need to prove that $\Omega_k(P_{G_1}) \subset \Omega_k(P_{G_2})$.

(a) We prove that
$$\Omega_k(P_{G_1}) \subset \mathcal{A}_k(P_{G_2})$$
$\forall \omega_k \in \Omega_k(P_{G_1})$, $\omega_k \in \mathcal{A}_k(P_{G_1})$, $\partial(\omega_k) \in \mathcal{A}_{k-1}(P_{G_1})$. Note that every $k$-path in $P_{G_1}$ is either an $k$-path in $P_{G_2}$ or starts with $v_1 v_2$, so $\omega_k$ can be represented as
$$\omega_k = v_1 v_2 e_{k-2} + e_k$$
where $e_{k-2} \in \mathcal{A}_{k-2}(P_{G_2})$ is a linear combination of $(k-2)$-paths $\sum v_{i_0} v_{i_1} \cdots v_{i_{k-2}}$ $(v_{i_0} \neq v_1)$ of $P_{G_2}$ and $e_k$ is an $k$-path of $P_{G_2}$. We have
$$\partial(\omega_k) = (v_2 - v_1)e_{k-2} + v_1 v_2 \partial(e_{k-2}) + \partial(e_k)$$
Note that $v_1 e_{k-2}$ is a linear combination of $(k-1)$-paths $v_1 v_{i_0} \cdots v_{i_{k-2}}$ $(v_1 \neq v_{i_0})$. Since $v_1 v_{i_0}$ is not an edge, these paths are not contained in $P_{G_1}$, but $\partial(w_k) \in \mathcal{A}_{k-1}(P_{G_1})$, so $v_1 e_{k-2}$ must add some item in the right part to become zero. There is not any item in the right part of the equation has $v_1 v_{i_0} \cdots v_{i_{k-2}}$, so $v_1 e_{k-2}$ must be zero, which means that $e_{k-2}$ is zero. Consequently,
$$\omega_k = e_k \in \mathcal{A}_k(P_{G_2})$$

(b) We prove that
$$\Omega_k(P_{G_2}) = \mathcal{A}_k(P_{G_2}) \cap \Omega_k(P_{G_1})$$
It is obvious that $\Omega_k(P_{G_2}) \subset \mathcal{A}_k(P_{G_2}) \cap \Omega_k(P_{G_1})$, so we only need to prove that $\mathcal{A}_k(P_{G_2}) \cap \Omega_k(P_{G_1}) \subset \Omega_k(P_{G_2})$.
$\forall e \in \mathcal{A}_k(P_{G_2}) \cap \Omega_k(P_{G_1})$, Since $e \in \mathcal{A}_k(P_{G_2})$, $v_1$ will not appear in $e$, which means $e$ is a path of $P_{G_2}$. Note that $e \in \Omega_k(P_{G_1})$, so $\partial(e) \in \mathcal{A}_{k-1}(P_{G_1})$, with the property that $e \in P_{G_2}$, we have $\partial(e) \in \mathcal{A}_{k-1}(P_{G_2})$, which means
$$e \in \Omega_k(P_{G_2})$$

Combining the results of (a) and (b), we have
$$\Omega_k(P_{G_1}) \subset \mathcal{A}_k(P_{G_2}) \cap \Omega_k(P_{G_1}) = \Omega_k(P_{G_2})$$

$\square$

Theorem B.16 means the path complex homology is a graph weak isomorphism invariant. Consequently, for two graphs $G_1$ and $G_2$, if there exists $k$ such that $\mathrm{H}_k(P_{G_1}) \ncong \mathrm{H}_k(P_{G_2})$, then $G_1$ and $G_2$ are not weak isomorphic and not isomorphic.

Figure 7 illustrates an example of the graph weak isomorphism. As shown in Figure 1, $G_1 = (V_1, E_1)$ where $V_1 = \{0, 1, 2, 3, 4, 5\}$, $E_1 = \{(0,1), (1,2), (2,3), (3,4), (4,5), (0,5)\}$. $G_2 = (V_2, E_2)$ where $V_2 = \{0, 1, 2, 3, 4, 5, 6, 7, 8\}$ and $E_2 = \{(0,1), (1,2), (2,3), (3,4), (4,5), (0,5), (5,6), (1,7), (3,8)\}$. $G_1$ can be derived by doing the graph collapse operation on $G_2$ through $\{7, (1,7)\}$, $\{6, (5,6)\}$ and $\{8, (3,8)\}$ one by one, so $G_1$ and $G_2$ are weak isomorphic. It can be seen that $G_1$ and $G_2$ both have one cycle and one connected component.

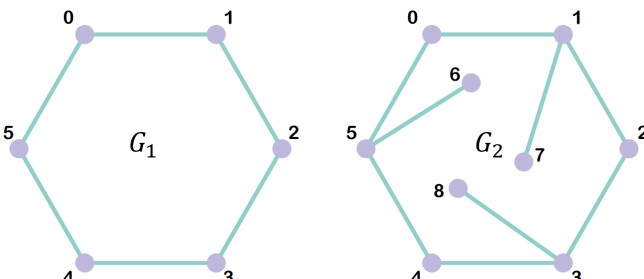

Figure 7: Illustration of the graph weak isomorphism. $G_1$ can be derived by doing the graph collapse operations on $G_2$ through $\{7, (1, 7)\}, \{6, (5, 6)\}$ and $\{8, (3, 8)\}$ one by one, so $G_1$ and $G_2$ are weak isomorphic.

## C  PATH COMPLEX INVARIANCE

Molecules can be effectively represented as graphs $G = (V, E)$, where vertices $V$ correspond to atoms, and edges $E$ represent chemical bonds. Building on this, a path complex $P_G$ is constructed from the graph, extending the connectivity into higher-dimensional simplices based on paths in $G$. These simplices encapsulate relationships such as chemical bond types and angular constraints between bonds, further enriching the molecular representation.

**Definition C.1** (Permutation Operation). A permutation $\pi$ is a bijection on the set of vertices $V$, representing a relabeling of the atoms in the molecule. This permutation induces a corresponding bijection on the path complex $P_G$, noted as $\pi(P_G)$, which preserves the connectivity structure of the original path complex.

**Definition C.2** (Graph Neural Network Layer for Path Complexes). Consider a generic layer of a graph neural network adapted for path complexes, updating the features of each simplex based on a message passing scheme:

$$h_{\sigma_n}^{(k+1)} = \text{RELU}\left( W^{(k)} \cdot h_{\sigma_n}^{(k)} + \sum_{\tau_n \in \mathcal{N}(\sigma_n)} U^{(k)} \cdot h_{\tau_n}^{(k)} \right)$$

$h_{\sigma_n}^{(k)}$ denotes the feature vector of the n-order path $\sigma_n$ at layer $k$. $W^{(k)}$ and $U^{(k)}$ are learnable parameters of the network at layer $k$. $\mathcal{N}(\sigma_n)$ represents the neighborhood set of simplex $\sigma_n$ in $P_G$, including faces and cofaces.

*Proof of Theorem B.5.*    1. Initially, let $h_{\sigma_n}^{(0)} = h_{\pi(\sigma_n)}^{(0)}$ for all $\sigma_n \in P_G$, assuming that the initial features are assigned consistently regardless of the labeling of the paths.

2. Assume that $h_{\sigma_n}^{(k)} = h_{\pi(\sigma_n)}^{(k)}$ holds true for some layer $k$.

3. To prove for layer $k + 1$:

$$h_{\sigma_n}^{(k+1)} = \text{RELU}\left( W^{(k)} \cdot h_{\sigma_n}^{(k)} + \sum_{\tau_n \in \mathcal{N}(\sigma_n)} U^{(k)} \cdot h_{\tau_n}^{(k)} \right)$$

$$h_{\pi(\sigma_n)}^{(k+1)} = \text{RELU}\left( W^{(k)} \cdot h_{\pi(\sigma_n)}^{(k)} + \sum_{\pi(\tau_n) \in \mathcal{N}(\pi(\sigma_n))} U^{(k)} \cdot h_{\pi(\tau_n)}^{(k)} \right)$$

Given the inductive hypothesis that $h_{\sigma_n}^{(k)} = h_{\pi(\sigma_n)}^{(k)}$ and $h_{\tau_n}^{(k)} = h_{\pi(\tau_n)}^{(k)}$ for all $\tau_n \in \mathcal{N}(\sigma_n)$, and recognizing that $\mathcal{N}(\pi(\sigma_n)) = \pi(\mathcal{N}(\sigma_n))$, it follows that:

$$h^{(k+1)}_{\sigma_n} = h^{(k+1)}_{\pi(\sigma_n)}$$

$\square$

# D PATH WEISFEILER LEHMAN (PWL) TEST

## D.1 PATH COMPLEX

**Definition D.1** (Path Complex Isomorphism). Given two path complexes $P_1, P_2$ over the vertices $V_1, V_2$. $P_1$ and $P_2$ are called isomorphic if there is a map $f : V_1 \to V_2$ such that $\sigma_n = v_0 v_1 \cdots v_n \in P_1 \iff f(\sigma) = f(v_0)f(v_1)\cdots f(v_n) \in P_2$.

**Theorem D.2.** *Given two graphs $G_1, G_2$, let $P_{G_1}, P_{G_2}$ be the path complexes derived from $G_1, G_2$ respectively. We have*

$$G_1 \cong G_2 \iff P_{G_1} \cong P_{G_2}$$

## D.2 PATH COMPLEX COLORING

**Definition D.3** (Path Coloring). A path coloring is a map $c$ such that for each path complex $P$ and any path $\sigma$ of $P$, $c(\sigma)$ is a color from a fixed color table. We denote this color by $c^P_\sigma$.

We will often omit $P$ in the subscript when the underlying path complex is arbitrary.

**Definition D.4.** Given two path complexes $P_1, P_2$ and a path coloring $c$. $P_1$ and $P_2$ are called $c$-similar, denoted by $c^{P_1} = c^{P_2}$, if for any dimension $n$, we have the color multi-sets equality

$$\{\{c^{P_1}_\sigma | dim(\sigma) = n, \sigma \in P_1\}\} = \{\{c^{P_2}_\tau | dim(\tau) = n, \tau \in P_2\}\}$$

**Definition D.5** (PWL). We give a path complex version of the WL test to derive a message passing procedure that can retain the expressive power of the test. We call this the Path WL (PWL), the steps of general PWL are as follows:

1. Given a path complex $P$, all the paths of $P$ are initialized with the same color.

2. For the color $c^t_\sigma$ of path $\sigma$ at iteration $t$, the color $c^{t+1}_\sigma$ of $\sigma$ at the next iteration is computed by perfectly hashing the color multi-set of the neighbors of $\sigma$.

3. The algorithm stops once a stable coloring is reached. Two path complexes are considered non-isomorphic if their color histograms are different at some dimensions.

**Neighbor Color Multi-set** Based on the four neighbor definitions, we have four types of neighbor color multi-sets. Let $c^t$ be the coloring of PWL for path complex $P$ at iteration $t$, four types of color multi-sets are as follows

1. $c^t_{\mathcal{B}}(\sigma) = \{\{c^t_\tau | \tau \in \mathcal{B}(\sigma)\}\}$
2. $c^t_{\mathcal{C}}(\sigma) = \{\{c^t_\tau | \tau \in \mathcal{C}(\sigma)\}$
3. $c^t_\uparrow(\sigma) = \{\{(c^t_\tau, c^t_{\sigma \cup \tau}) | \tau \in \mathcal{N}_\uparrow(\sigma)\}$
4. $c^t_\downarrow(\sigma) = \{\{(c^t_\tau, c^t_{\sigma \cap \tau}) | \tau \in \mathcal{N}_\downarrow(\sigma)\}$

Having the neighbor color multi-sets, we obtain the following update rule that contains all four types of neighbors:

$$c^{t+1}_\sigma = \text{HASH}\{c^t_\sigma, c^t_{\mathcal{B}}(\sigma), c^t_{\mathcal{C}}(\sigma), c^t_\uparrow(\sigma), c^t_\downarrow(\sigma)\}$$

Actually, certain neighbors can be removed without affecting the expressive power of PWL test in terms of path complex that can be differentiated.

**Theorem D.6.** *PWL with $\text{HASH}\{c^t_\sigma, c^t_{\mathcal{B}}(\sigma), c^t_\uparrow(\sigma)\}$ is as powerful as PWL with the four-neighbor-updating strategy $\text{HASH}\{c^t_\sigma, c^t_{\mathcal{B}}(\sigma), c^t_{\mathcal{C}}(\sigma), c^t_\uparrow(\sigma), c^t_\downarrow(\sigma)\}$.*

**Theorem D.7.** *PWL is strictly more powerful than WL.*

**Theorem D.8.** *PWL is no less powerful than SWL Bodnar et al. (2021b) with the clique complex lifting.*

### D.3 PATH COMPLEX MESSAGE PASSING

We propose a general Path Complex Message Passing (PCMP) using the following messages passing operations. For a path $\sigma$ in $P$, we have

$$m_{\mathcal{B}}^{t+1}(\sigma) = AGG_{\tau \in \mathcal{B}(\sigma)}(M_{\mathcal{B}}(h_\sigma^t, h_\tau^t)) \tag{4}$$

$$m_\uparrow^{t+1}(\sigma) = AGG_{\tau \in \mathcal{N}_\uparrow(\sigma)}(M_\uparrow(h_\sigma^t, h_\tau^t, h_{\sigma \cup \tau}^t)) \tag{5}$$

Then, the updating function considers these two types of messages and the previous color of $\sigma$:

$$h^{t+1}(\sigma) = U(h_\sigma^t, m_{\mathcal{B}}^t(\sigma), m_\uparrow^t(\sigma)) \tag{6}$$

After L layers of the message passing process, the readout function takes the color multi-sets at all dimensions as input:

$$h_P = \text{READOUT}(\{\{h_\sigma^L\}\}_{dim(\sigma)=0}, \cdots, \{\{h_\tau^L\}\}_{dim(\tau)=p}) \tag{7}$$

**Theorem D.9.** *PCMP with sufficient layers and injective neighborhood aggregators are as powerful as PWL.*

### D.4 PROOF OF MAIN RESULTS

In order to prove the main results, we give some notations.

**Definition D.10** (Path Coloring Refinement). A path coloring $c$ refines a path coloring $d$, denoted by $c \sqsubseteq d$, if for any path complex $P_1, P_2$ and $\sigma \in P_1$, $\tau \in P_2$, $c_\sigma^{P_1} = c_\tau^{P_2}$ implies $d_\sigma^{P_1} = d_\tau^{P_2}$. Additionally, if $d \sqsubseteq c$, we say that $c$ and $d$ are equivalent.

**Lemma D.11.** *Given two path complexes $P_1, P_2$ with $A \subset P_1$, $B \subset P_2$. Assume $c$ and $d$ are two path coloring such that $c \sqsubseteq d$. If $\{\{d_\sigma^{P_1} | \sigma \in A\}\} \neq \{\{d_\tau^{P_2} | \tau \in B\}\}$, then $\{\{c_\sigma^{P_1} | \sigma \in A\}\} \neq \{\{c_\tau^{P_2} | \tau \in B\}\}$.*

*Proof.* Let $C_1 = \{\{c_\sigma^{P_1} | \sigma \in A\}\}$, $C_2 = \{\{c_\tau^{P_2} | \tau \in B\}\}$. Assume $C_1 = C_2$, then there is a bijection $f : A \to B$ such that $\forall \sigma \in A$, $\tau = f(\sigma)$, we have $c_\sigma^{P_1} = c_\tau^{P_2}$. From $c \sqsubseteq d$ we know $d_\sigma^{P_1} = d_\tau^{P_2}$. Consequently, $\{\{d_\sigma^{P_1} | \sigma \in A\}\} = \{\{d_{f(\sigma)}^{P_2} | \sigma \in A\}\} = \{\{d_\tau^{P_2} | \tau \in B\}\}$, which contradicts with the condition that $\{\{d_\sigma^{P_1} | \sigma \in A\}\} \neq \{\{d_\tau^{P_2} | \tau \in B\}\}$. Hence the assumption is wrong. $\square$

**Corollary D.12.** *Given two path colorings $c$ and $d$ such that $c \sqsubseteq d$. If $d^{P_1} \neq d^{P_2}$, then $c^{P_1} \neq c^{P_2}$.*

*Proof.* This follows by replacing the subsets $A, B$ by the sets of $n$-paths of $P_1$ and $P_2$ respectively in the proof of Lemma D.11. $\square$

The above corollary D.12 means that if $c$ refines $d$, then $c$ is able to distinguish all the path complex pairs that $d$ can distinguish. In this sense, we can say that $c$ is at least as powerful as $d$. If $c$ and $d$ are equivalent, we say they have the same expressive power.

*Proof of Theorem D.2.* It is easy to see that if $G_1 \cong G_2$, then $P_{G_1} \cong P_{G_2}$. The inverse statement follows from the fact that any graph is a subcomplex of its derived path complex by considering the 0-paths and 1-paths. $\square$

*Proof of Theorem D.6.* Let $a^t$ be the coloring at iteration t of the updating startegy

$$\text{HASH}\{a_\sigma^t, a_{\mathcal{B}}^t(\sigma), a_{\mathcal{C}}^t(\sigma), a_\uparrow^t(\sigma), a_\downarrow^t(\sigma)\}$$

$b^t$ be the coloring at iteration t of the updating strategy

$$\text{HASH}\{b_\sigma^t, b_{\mathcal{B}}^t(\sigma), b_\uparrow^t(\sigma), b_\downarrow^t(\sigma)\}$$

$c^t$ be the coloring at iteration t of the updating strategy

$$\text{HASH}\{c_\sigma^t, c_{\mathcal{B}}^t(\sigma), c_\uparrow^t(\sigma)\}$$

We firstly prove that $a^t$ and $b^t$ are equivalent, then prove that $b^t$ and $c^t$ are equivalent.

1. $a^t$ and $b^t$ are equivalent. We have $a^t \sqsubseteq b^t$ because $a^t$ contains additional colors of its coface neighbors in the color updating rule. It suffices to prove that $b^t \sqsubseteq a^t$. We do this by induction. The base case holds since all the paths are initialized with the same color. Assume the result holds for $t = k$, we prove that $b^{k+1} \sqsubseteq a^{k+1}$. Let $\sigma \in P_1$ and $\tau \in P_2$ be two $n$-paths from two arbitrary path complexes, suppose $b_\sigma^{k+1} = b_\tau^{k+1}$, we prove that $a_\sigma^{k+1} = a_\tau^{k+1}$.

   The equation $b_\sigma^{k+1} = b_\tau^{k+1}$ means that the hash function at iteration $t+1$ have the same arguments. Consequently, $b_\sigma^k = b_\tau^k$, $b_\mathcal{B}^k(\sigma) = b_\mathcal{B}^k(\tau)$, $b_\uparrow^k(\sigma) = b_\uparrow^k(\tau)$, $b_\downarrow^k(\sigma) = b_\downarrow^k(\tau)$. We prove that $b_\mathcal{C}^k(\sigma) = b_\mathcal{C}^k(\tau)$.

   We have $b_\uparrow^k(\sigma) = b_\uparrow^k(\tau)$ and

   $$b_\uparrow^k(\sigma) = \{\!\{(b_e^k, b_{\sigma \cup e}^k)|e \in \mathcal{N}_\uparrow(\sigma)\}\!\}, b_\uparrow^k(\tau) = \{\!\{(b_e^k, b_{\tau \cup e}^k)|e \in \mathcal{N}_\uparrow(\tau)\}\!\} \tag{8}$$

   Replacing the first component of the tuple by the same color, we have

   $$\{\!\{(-, b_{\sigma \cup e}^k)|e \in \mathcal{N}_\uparrow(\sigma)\}\!\} = \{\!\{(-, b_{\tau \cup e}^k)|e \in \mathcal{N}_\uparrow(\tau)\}\!\} \tag{9}$$

   By the definition of upper adjacency and coface we have

   $$b_\mathcal{C}^k(\sigma) = \{\!\{b_w^k|w \in \mathcal{C}(\sigma)\}\!\} = \{\!\{b_{\sigma \cup e}^k|e \in \mathcal{N}_\uparrow(\sigma)\}\!\} \tag{10}$$

   $$b_\mathcal{C}^k(\tau) = \{\!\{b_w^k|w \in \mathcal{C}(\tau)\}\!\} = \{\!\{b_{\tau \cup e}^k|e \in \mathcal{N}_\uparrow(\tau)\}\!\} \tag{11}$$

   Combining Equation (8), (9), (10), (11), we have $b_\mathcal{C}^k(\sigma) = b_\mathcal{C}^k(\tau)$.

   From the induction hypothesis, we have $a_\sigma^k = a_\tau^k$, $a_\mathcal{B}^k(\sigma) = a_\mathcal{B}^k(\tau)$, $a_\mathcal{C}^k(\sigma) = a_\mathcal{C}^k(\tau)$, $a_\uparrow^k(\sigma) = a_\uparrow^k(\tau)$, $a_\downarrow^k(\sigma) = a_\downarrow^k(\tau)$, so $a_\sigma^{k+1} = a_\tau^{k+1}$.

2. $b^t$ and $c^t$ are equivalent. Similarly we have $b^t \sqsubseteq c^t$, we further prove that $c^{2t} \sqsubseteq b^t$. We do this by induction. The base case is obvious because all the paths are initialized with the same color. Assume the results holds for $t = k$, we prove that $c^{2k+2} \sqsubseteq b^{k+1}$. Let $\sigma \in P_1$ and $\tau \in P_2$ be two $n$-paths from two arbitrary path complexes, suppose $c_\sigma^{2k+2} = c_\tau^{2k+2}$, we prove that $b_\sigma^{k+1} = b_\tau^{k+1}$.

   For $c_\sigma^{2k+2} = c_\tau^{2k+2}$, by going back two steps of the hash function, we have $c_\sigma^{2k} = c_\tau^{2k}$, $c_\mathcal{B}^{2k}(\sigma) = c_\mathcal{B}^{2k}(\tau)$, $c_\uparrow^{2k}(\sigma) = c_\uparrow^{2k}(\tau)$. We want to prove that $c_\downarrow^{2k}(\sigma) = c_\downarrow^{2k}(\tau)$.

   Assume $c_\downarrow^{2k}(\sigma) \neq c_\downarrow^{2k}(\tau)$, then there is a color pair $(c_0, c_1)$ such that $(c_0, c_1)$ appears more times in $c_\downarrow^{2k}(\sigma)$ (without loss of generality) than in $c_\downarrow^{2k}(\tau)$. For any path $\delta$ and $\lambda$, define

   $$A(\delta) = \{\!\{(c_\phi^{2k} = c_0, c_\delta^{2k} = c_1)|\phi \in \mathcal{C}(\delta)\}\!\} \tag{12}$$

   $$C_\lambda = \{\!\{|A(\delta)||\delta \in \mathcal{B}(\lambda)\}\!\} \tag{13}$$

   Then we have

   $$C_\sigma = \{\!\{|A(\delta)||\delta \in \mathcal{B}(\sigma)\}\!\} = \{\!\{|(c_\phi^{2k} = c_0, c_\delta^{2k} = c_1)||\delta \in \phi \cap \sigma\}\!\} \tag{14}$$

   $$C_\tau = \{\!\{|A(\delta)||\delta \in \mathcal{B}(\tau)\}\!\} = \{\!\{|(c_\phi^{2k} = c_0, c_\delta^{2k} = c_1)||\delta \in \phi \cap \tau\}\!\} \tag{15}$$

   So $C_\sigma \neq C_\tau$.

   Considering the path coloring $d(\delta) = |A(\delta)|$. For two $n$-paths $\delta_1, \delta_2$, if $d(\delta_1) \neq d(\delta_2)$, we can assume that $|A(\delta_1)| > |A(\delta_2)|$ without loss of generality, then the number of upper adjacent neighbors of $\delta_1$ and $\delta_2$ up to color pair $(c_0, c_1)$ are different, which means $c_\uparrow^{2k}(\delta_1) \neq c_\uparrow^{2k}(\delta_2)$. So $c_{\delta_1}^{2k+1} \neq c_{\delta_2}^{2k+1}$, which means $c^{2k+1} \sqsubseteq d$.

   Applying Lemma D.11 to $\mathcal{B}(\sigma)$ and $\mathcal{B}(\tau)$, we have

   $$\{\!\{c_{\delta_1}^{2k+1}|\delta_1 \in \mathcal{B}(\sigma)\}\!\} \neq \{\!\{c_{\delta_2}^{2k+1}|\delta_2 \in \mathcal{B}(\tau)\}\!\} \tag{16}$$

   The above multi-sets are exactly the color multi-sets of the faces of $\sigma$ and $\tau$, which means $c_\mathcal{B}^{2k+1}(\sigma) \neq c_\mathcal{B}^{2k+1}(\tau)$. Consequently, $c_\sigma^{2k+2} \neq c_\tau^{2k+2}$, which contradicts with the induction hypothesis, so $c_\downarrow^{2k}(\sigma) = c_\downarrow^{2k}(\tau)$.

   From the induction hypothesis, we have $b_\sigma^k = b_\tau^k$, $b_\mathcal{B}^k(\sigma) = b_\mathcal{B}^k(\tau)$, $b_\uparrow^k(\sigma) = b_\uparrow^k(\tau)$, $b_\downarrow^k(\sigma) = b_\downarrow^k(\tau)$, so $b_\sigma^{k+1} = b_\tau^{k+1}$.

□

*Proof of Theorem D.7.* Given a path complex $P$, let $a^t$ be the coloring of the vertices of $P$ at iteration t of WL and $b^t$ be the coloring of the same vertices at iteration t of PWL. We firstly prove that $b^t \sqsubseteq a^t$, then give a pair of graphs to show that they cannot be differentiated by WL but can be differentiated by PWL.

1. $b^t \sqsubseteq a^t$. We do this by induction. The base case holds because all vertices are initialized with the same color. Suppose the result holds for $t = k$, we prove that $b^{k+1} \sqsubseteq a^{k+1}$. Let $v$ and $w$ be two vertices of two arbitrary path complexes $P_1, P_2$, suppose $b_v^{k+1} = b_w^{k+1}$, we prove that $a_v^{k+1} = a_w^{k+1}$.

   Note that vertices only has upper adjacent neighbors, so we have $b_v^k = b_w^k, b_\uparrow^k(v) = b_\uparrow^k(w)$. The second equation means

   $$\{\{b_x^k | (b_x^k, -) \in b_\uparrow^k(v)\}\} = \{\{b_y^k | (b_y^k, -) \in b_\uparrow^k(w)\}\}$$

   This can be equivalently written as

   $$\{\{b_x^k | x \in \mathcal{N}_\uparrow(v)\}\} = \{\{b_y^k | y \in \mathcal{N}_\uparrow(w)\}\}$$

   From the induction hypothesis, we have $a_v^k = a_w^k$ and

   $$\{\{a_x^k | x \in \mathcal{N}_\uparrow(v)\}\} = \{\{a_y^k | y \in \mathcal{N}_\uparrow(w)\}\}$$

   These are the arguments of the hash function for WL to compute the colors of $v$ and $w$ in the next iteration, so $a_v^{k+1} = a_w^{k+1}$.

2. Considering the graphs in Figure 8, they cannot be differentiated by WL test. In PWL test, the path complex derived from the right graph has not any 3-path while the derived path complex from the left graph has 3-paths.

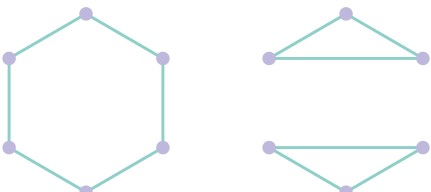

Figure 8: Two graphs that cannot be distinguished by WL but can be differentiated by PWL.

□

*Proof of Theorem D.8.* Considering the graphs in Figure 9, they cannot be differentiated by SWL test. In PWL test, the path complex derived from the right graph has not any 4-path while the derived path complex from the left graph has 4-paths. □

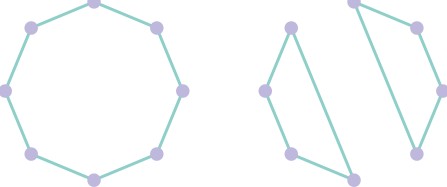

Figure 9: Two graphs that cannot be distinguished by SWL but can be differentiated by PWL.

*Proof of Theorem D.9.* Let $b^t$ and $d^t$ be the coloring at iteration t of PWL and the t-th layer of an PCMP respectively. Assume the PCMP has L layers and assume $d^t = d^L (t > L)$. We use induction to prove that $d^t \sqsubseteq b^t$. The base case holds by definition. Suppose the result holds for $t = k$, when $t = k + 1$, we prove that $d^{k+1} \sqsubseteq b^{k+1}$. For any two $n$-paths $\sigma, \tau$ of any two path complexes $P_1, P_2$ such that $d_\sigma^{k+1} = d_\tau^{k+1}$, we prove that $b_\sigma^{k+1} = b_\tau^{k+1}$.

The condition means all the update, aggregate and message functions are injective, so their composition is also injective. Hence $d_\sigma^k = d_\tau^k, d_\mathcal{B}^k(\sigma) = d_\mathcal{B}^k(\tau), d_\uparrow^k(\sigma) = d_\uparrow^k(\tau)$.

$d_\mathcal{B}^k(\sigma) = d_\mathcal{B}^k(\tau)$ means

$$\{\{d_s^k | s \in \mathcal{B}(\sigma)\}\} = \{\{d_t^k | t \in \mathcal{B}(\tau)\}\}$$

$d_\uparrow^k(\sigma) = d_\uparrow^k(\tau)$ means

$$\{\{(d_s^k, d_{s\cup\sigma}^k) | s \in \mathcal{N}_\uparrow(\sigma)\}\} = \{\{(d_t^k, d_{t\cup\tau}^k) | t \in \mathcal{N}_\uparrow(\tau)\}\}$$

By the induction hypothesis, we have $b_\sigma^k = b_\tau^k$.

$$\{\{b_s^k | s \in \mathcal{B}(\sigma)\}\} = \{\{b_t^k | t \in \mathcal{B}(\tau)\}\}$$

$$\{\{(b_s^k, b_{s\cup\sigma}^k) | s \in \mathcal{N}_\uparrow(\sigma)\}\} = \{\{(b_t^k, b_{t\cup\tau}^k) | t \in \mathcal{N}_\uparrow(\tau)\}\}$$

So $b_\sigma^k = b_\tau^k, b_\mathcal{B}^k(\sigma) = b_\mathcal{B}^k(\tau), b_\uparrow^k(\sigma) = b_\uparrow^k(\tau)$, these are the arguments of the hash function in PWL, so $b_\sigma^{k+1} = b_\tau^{k+1}$.

$\square$

