# OpenReview forum: "Path Complex Message Passing for Molecular Property Prediction"
_ICLR.cc/2025/Conference — Submitted to ICLR 2025_

### Official Review · Reviewer_CZzj · 2024-10-27

**Soundness:** 2
**Presentation:** 2
**Contribution:** 2
**Rating:** 5
**Confidence:** 3

**Summary:**

This paper introduces Path Complex Message Passing (PCMP), a method for molecular property prediction that represents molecules using path complexes of different orders to capture various aspects of molecular structure (bond lengths, angles, and dihedral angles). The work is heavily theoretical. The method is tested on five subsets of MoleculeNet, showing improvements over baseline models. Most of the papers content is theoretical development and proofs, with a relatively brief experimental section.

**Strengths:**

The paper displays technical rigor in developing its mathematical foundations, establishing formal proofs for path complex properties and their relationship to molecular structures. The proposed path complex representation is well motivated from a chemistry perspective, making explicit connections to molecular force fields and showing how different path orders correspond to physical properties (bond lengths, angles, and dihedral angles). The architecture is novel. While limited, the experimental results do show consistent improvements across multiple benchmark datasets, and the ablation studies help in distilling the contribution of different components of the model.

**Weaknesses:**

The paper suffers from several weaknesses. In the Introduction, the authors refer to quite 'old' papers. The experimental validation is notably thin compared to the extensive theoretical development, taking up only about 2 pages of the 10-page paper. The empirical improvements, while consistent, are relatively modest and don't seem to justify the substantial complexity introduced by the method. There's inadequate discussion of computational overhead and scalability. The path complex representation likely introduces significant computational costs, but this isn't analyzed. Even though datasets are described in the appendix, authors haven't included any brief description of the dataset, or the task at hand: for example, for QM9, one cannot know against which property the authors are regressing against from the main paper.

**Questions:**

1) What is the time and memory complexity of constructing and operating on path complexes compared to traditional graph-based methods?  How does the method scale with molecule size?

2) Related to above. Could you provide empirical justification for using 3-path complexes as the maximum order? What happens with higher orders?

3) Could you provide a brief description of each dataset's characteristics in the main paper?

---

> ### Author Response · Authors · 2024-11-22
> **Response to Reviewer CZzj**
>
> Thank you for your constructive feedback and comments, which have greatly helped us improve the clarity and depth of our manuscript.
> ## Weaknesses
>
> ### Weakness 1:**_Older papers_**
>
> Answer 1: Our baseline methods include several recent approaches, such as DMP by Zhu et al. (2023), SMPT by Li et al. (2024), and DGCL by Jiang et al. (2024). These references highlight our effort to incorporate state-of-the-art methods into our analysis, ensuring that our comparisons are both relevant and up-to-date.
>
> ### Weakness 2: **_More theoretical than experimental_**
>
> Answer 2: We appreciate your observation regarding the balance between theoretical development and experimental validation. In response, we have included visualizations of different path orders and performance comparisons of path complexes with various geometric features (Pages 9 and 10).
>
> Our initial focus was to establish the theoretical foundation of the PCMP method, introducing a novel topological framework for molecular representation. This framework utilizes topology to depict molecular structures, serving as the basis for the message passing and pooling mechanisms. The results from our regression and classification tasks, along with ablation studies, demonstrate the practical utility and predictive efficacy of path complexes in molecular property prediction.
>
> ### Weakness 3: **_Computational expense_**
>
> Answer 3: The PCMP method constructs higher-order paths using only covalent bonds, which significantly reduces computational complexity compared to traditional methods that include both covalent and non-covalent bonds. This strategic design not only retains the model’s effectiveness but also optimizes the use of computational resources, achieving a balance between performance and resource demands.
>
> To further enhance efficiency, the computational overhead of path complexes can be reduced by minimizing the use of high-order paths. This ensures scalability while maintaining the accuracy of predictions.
>
>
> ### Weakness 4: **_Dataset description and tasks_**
>
> Answer 4: In Appendix A.3, we provide a detailed introduction to our datasets, particularly regarding the predictive targets for the QM9 dataset. In our revision, we include concise descriptions of each dataset and elaborate on the specific molecular properties we aim to predict, such as the electron spatial extent (α) and the energies of the highest occupied molecular orbital (εHOMO) and the lowest unoccupied molecular orbital (εLUMO).
>
>
> ## Questions
> ### Question 1: **_Time and memory complexity_**
>
> Answer 1: In the PCMP method, higher-order path construction is restricted to covalent bonds, significantly reducing the computational burden compared to approaches that consider all interactions, including non-covalent ones. The computational complexity of our algorithm is O(N^2), where N represents the number of atoms in a molecule. This quadratic complexity primarily arises from pairwise interactions in path complex formation. However, by limiting interactions to covalent bonds, the number of processed interactions remains manageable, ensuring computational efficiency.
>
> ### Question2: **_Using 3-path complexes, higher orders path_**?
>
> Answer2: Our approach considers up to three-body (3-path) interactions, effectively capturing interactions among four atoms. This design parallels considerations of dihedral and improper angles in molecular force fields, providing a robust and comprehensive representation of molecular structure. As higher-order paths introduce additional complexity, designing effective geometric features for these paths is a key focus of ongoing optimization in our model.
>
> ### Question 3:  **_Dataset_**
> Answer3:  The QM7 and QM9 datasets are related to quantum mechanics, while the Tox21, HIV, and MUV datasets pertain to physiology and biophysics. For comprehensive details about the datasets, please refer to Appendix A.3.

---

### Official Review · Reviewer_bvsi · 2024-10-30

**Soundness:** 1
**Presentation:** 2
**Contribution:** 1
**Rating:** 5
**Confidence:** 5

**Summary:**

The paper presents **Path Complex Message Passing (PCMP)**, which introduces path complexes to model intricate chemical and non-chemical interactions for molecular property prediction. The model demonstrates promising results on molecular benchmark datasets and aims to provide a more detailed molecular representation by incorporating high-order interactions in a path complex framework.

**Strengths:**

1. **Innovative Modeling of Molecular Interactions**: PCMP introduces path complexes, offering an approach to capture both local and global molecular features that go beyond traditional graph representations.
2. **Methodological Rigor**: The hierarchical message-passing mechanism is well-detailed, showing how path complexes of different orders contribute to the molecular representation.
3. **Thorough Ablation Studies**: The authors provide in-depth ablation studies that highlight the importance of various path orders and message-passing mechanisms, strengthening the evaluation.

**Weaknesses:**

1. **Unsubstantiated Claim on Force Field Mimicking**: A major claim of the paper is that PCMP mimics the molecular mechanics (MM) force field, yet no benchmarks or empirical results using MM force field datasets (e.g., MD17, MD22) are provided to substantiate this claim. Without benchmarking against MM force fields, the claim appears unsupported, and this oversight detracts from the paper's validity in this area.

2. **Computational Complexity**: The inclusion of path complexes, especially higher-order ones, is likely computationally demanding. However, the authors do not provide insights into potential trade-offs, such as runtime or scalability on larger datasets.

3. **Lack of Equivariance in Model Design**: Given the model’s target application in molecular property prediction, its architecture does not incorporate rotational or translational equivariance, which would enhance its ability to handle spatial molecular data more robustly. Adding equivariant layers could make the model better suited to capturing geometry-sensitive properties.

4. **Interpretability**: The model’s complex hierarchical structure might hinder interpretability, as it’s not clear which paths contribute most significantly to predictions or whether high-order interactions have consistent relevance across datasets. A comprehensive interpretability study is recommended.

**Questions:**

1. Can the authors provide empirical validation on MM force field datasets to substantiate their claim of mimicking MM force fields?
2. Are there specific path orders or features that consistently contribute more to accurate predictions? Could this be visualized or quantified?
3.  Can the model be used to capture the improper angle? Is it possible to implement this?
4.  For Figure 4, does the path complex graph alleviate the over-squashing? If so, can the author provide an empirical study on this, for example, compare the effective resistance.
5.  So the current experiments contain only validation from small molecules dataset, what about for large molecules? How does the model scale to larger molecules?

---

> ### Author Response · Authors · 2024-11-22
> **Response to Reviewer bvsi**
>
> Thank you for your thoughtful feedback and questions, which have provided valuable insights to improve our manuscript.
> ## Weakness
>
> ### Weakness1:  **_Unsubstantiated Claim on Force Field Mimicking._**
>
> Answer1: We appreciate the opportunity to clarify this point. **Our primary objective is not to predict molecular dynamics (MD) processes directly. Instead, we draw inspiration from MD principles, leveraging geometric features derived from atomic positions to predict molecular properties.** Since our model does not use atomic coordinates as inputs, it cannot predict forces or derive partial derivatives of outputs with respect to coordinates. This design aligns our approach more closely with topological representations of molecular geometry than with direct MD simulations.
>
> ### Weakness 2: **_Computational Complexity._**
>
> Answer2: Thank you for addressing the computational aspects of our method. **In our PCMP approach, the construction of higher-order paths is restricted to covalent bonds. This choice significantly reduces the computational burden compared to methods that include all possible interactions, such as non-covalent bonds.**
>
> The computational complexity of our algorithm is \(O(N^2)\), where \(N\) is the number of atoms in a molecule. This quadratic complexity arises from pairwise interactions during the formation of path complexes. However, by limiting high-order paths to covalent bonds only, we effectively manage the number of interactions processed, ensuring that runtime remains feasible for practical applications.
>
> ### Weakness 3: **_Lack of Equivariance in Model Design_**
>
> Answer 3: Our model ensures invariance and equivariance by employing relative geometric features, such as angles and distances, which inherently account for variations due to rotations and translations. **This approach enables the model to effectively capture essential geometric dependencies required for accurate molecular property predictions without relying on absolute coordinates.** Furthermore, we explicitly state the invariance of the Path Complex in **Theorem 3.3** (Page 4) and provide a detailed proof in the Appendix.
>
> ### Weakness 4: **_Interpretability_**
>
> Answer 4: The interpretability of our model stems from its use of Path Complexes to capture molecular force fields and encode detailed geometric information. By incorporating higher-order geometric features, such as bond angles, dihedral angles. Path Complexes provide a structured and physically meaningful representation of molecular systems.
>
> To further enhance interpretability, we designed a bidirectional information transfer mechanism, allowing effective communication between high-order paths (e.g., 3-paths) and lower-order paths (e.g., 0-paths). The validity of this mechanism was demonstrated through ablation studies (Page 8), showing its contribution to improved predictive performance. Additionally, in the newly added section “Impact of Geometric Features in Path Complex” (Page 10), we empirically validated the effectiveness of integrating more geometric information.
>
> These results illustrate that the Path Complex framework not only improves performance but also provides insights into how molecular geometries influence property predictions, contributing to the interpretability of our approach.
>
> ## Questions: **_MM force fields, path orders or feature, over-squashing, large molecules_**
> 1. For MM force field, the answer is in General Response
> 2. For path orders or feature question. Due to the message passing mechanism in the path complex, updating the information for an n-order path necessitates the use of both n-1 order paths and n+1 order paths, thereby fixing the sequence of path orders within the path complex. **We added Figure 6 illustrates the influence of different path orders on regression and classification tasks. The section \paragraph{Impact of Geometric Feature in Path Complex}” (page 10) demonstrates how various path features affect the model’s performance.**
> 3. Our 3-path representation provides a framework for characterizing 4-body interactions, including improper angles. By explicitly incorporating features of improper angles into the 3-path configuration, we can effectively encode their geometric properties within our graph-based model. This approach allows us to capture the structural nuances of improper angles, enhancing the model’s ability to accurately represent molecular geometries.
> 4. The issue of over-squashing is a significant problem worthy of investigation within the context of Graph Neural Networks (GNNs). However, in this paper, we focus solely on exploring the expressive capabilities of PWL.
> 5. We acknowledge the limitation of not including macromolecular structures in our study, primarily due to the significant computational demands of applying our path complex method to large molecules. Currently, our resources are more suited to smaller molecules.

---

> > ### Comment · Reviewer_bvsi · 2024-11-23
> >
> > Thanks you for your comments. I’ll maintain my score.

---

### Official Review · Reviewer_mQvP · 2024-11-02

**Soundness:** 3
**Presentation:** 2
**Contribution:** 3
**Rating:** 6
**Confidence:** 2

**Summary:**

The paper introduces a novel approach called Path Complex Message Passing (PCMP) for molecular property prediction using geometric deep learning. Unlike traditional graph neural networks (GNNs) that operate on molecular graphs, PCMP employs path complexes that capture multi-body interactions in molecules through paths of various orders. The model’s hierarchical message-passing mechanism updates high-order paths first, followed by lower-order paths, facilitating effective feature communication between these paths. Extensive experiments on benchmark datasets demonstrate that PCMP achieves state-of-the-art results in molecular property prediction, showcasing its potential to model complex molecular interactions comprehensively.

**Strengths:**

* Originality: The PCMP model presents a unique innovation in molecular property prediction by applying path complexes, which go beyond conventional graph-based representations. The incorporation of multi-order path complexes allows for capturing high-order interactions like bond angles and dihedral angles. This approach offers a fresh perspective on molecular graph representation, making PCMP stand out from other GNN-based models that primarily rely on node and edge interactions.
* Quality: The paper provides a thorough experimental validation, comparing PCMP with a diverse set of baseline models, including both pretrained and non-pretrained GNNs.

**Weaknesses:**

* Clarity: I suggest the authors to hide some of the technical details in the Appendix.
* Experimental Limitations: Although the experiments are extensive, the paper could benefit from a more diverse set of benchmarks. The current datasets focus primarily on small to medium-sized molecules, while macromolecular structures, such as proteins, are absent from the evaluation. In addition, the test of efficiency, i.e. speed of training or inferences, is missing.

**Questions:**

Overall, I believe the authors present a good tool to predict molecular properties. However, the following questions should be addressed to clarify key aspects and improve rigor:

1. How do you compare the accuracy of your model with the more recent models, M3GNet, MACE, or EquiformerV2?

    - In addition, you only compared the accuracy, how about the speed compared with other models?

2. Is it possible to expand this framework to periodic systems, i.e. inorganic materials?

3. **Interpretability of Path Features**: Could the authors explore or comment on how path features across different orders contribute to the final prediction? Are there plans to visualize or interpret specific paths in relation to molecular properties?

---

> ### Author Response · Authors · 2024-11-22
> **Response to Reviewer mQvP**
>
> We appreciate your questions and suggestions, which have helped us refine and clarify our work.
>
> ## Weakness
>
> ### Weakness 1: **_Clarity_**
>
> Answer 1: ** We agree that relocating some of the more intricate details to an appendix would make the manuscript more accessible to general readers while providing additional depth for interested researchers. Accordingly, we will reorganize the manuscript to shift complex technical sections to an appendix, improving overall readability.
>
> ### Weakness 2: **_Experimental Limitations_**
>
> Answer 2: We acknowledge that a limitation of our study is the exclusion of large molecular structures, which stems from the challenges of designing geometric features based on protein positions. Currently, our focus and resources are geared towards smaller molecules. While our focus is on predicting properties of small molecules, our model still performs well on datasets like QM9, which has over a hundred thousand samples, and the MUV dataset, which is nearly as large.
> For our own testing, here are the specific details:
> 1. QM7 dataset: Running 500 epochs under our test parameters required approximately 2 hours for five iterations.
> 2. QM9 dataset: Due to the smaller batch size, it took about 12 hours on average to complete 500 epochs across five iterations.
> 3. Tox21 dataset: Completing 1000 epochs required an average of 4 hours over five iterations.
> 4. HIV dataset: Completing 1000 epochs required an average of 18 hours over five iterations.
> 5. MUV dataset: Completing 1000 epochs required an average of 39 hours over five iterations.
>
> ## Questions
>
> ### Question 1: **_Recently models, M3GNet, MACE, or EquiformerV2?_**
> Answer 1: Thank you for raising this point. Unlike models such as M3GNet, MACE, or EquiformerV2, which focus on molecular dynamics (MD) tasks, our approach is designed for molecular property prediction based on geometric features. Since we do not input atomic coordinates directly, our model cannot predict forces or derive partial derivatives of outputs with respect to coordinates. Instead, we leverage relative geometric features to capture molecular properties inspired by MD principles.
> ### Question 2: **_How about the speed compared with other models?_**
> While direct speed comparisons with other models are challenging due to limited reporting on computational overheads in prior works, we highlight key points regarding the efficiency of the PCMP model:
> 1. **No Pre-training Required**: Unlike many models requiring pre-training, PCMP skips this step, significantly reducing preparation time and enabling rapid deployment and iterative testing.
> 2. **Computational Overhead**: Compared to models like Mol-GDL (), which do not use pre-training, PCMP is approximately **twice as slow**, primarily due to the additional complexity of constructing and processing path complexes, particularly higher-order paths.
> 3. **Performance-Speed Trade-off**: The additional computational cost is justified by the model's ability to capture intricate molecular interactions and geometric features, as evidenced by its superior performance on multiple benchmark datasets.
>
> ### Question 3: **_Expand this framework to periodic systems, i.e. inorganic materials?_**
> Answer 3: Given the structural similarities between molecular materials and inorganic materials, we believe our PCMP can be adapted to periodic systems. From a topological perspective, in addition to path complexes, the quotient graph also presents a viable option. We are optimistic about using topological techniques to handle similar geometric datasets and plan to explore this direction in future research.
>
> ### Question 4: **_Interpretability of Path Features_**
> Answer 4: We have added a new section titled \paragraph{Impact of Geometric Feature in Path Complex} on page 10. Figure 6 visualizes the impact of path orders on regression and classification tasks. Furthermore, in the ablation study section, we analyzed how different path orders affect the model’s predictive accuracy. Our findings indicate that higher-order paths significantly enhance the model’s ability to predict molecular properties.

---

> > ### Comment · Reviewer_mQvP · 2024-11-27
> >
> > Thanks you for your comments. I keep my original recommendation.

---

### Official Review · Reviewer_hgFA · 2024-11-02

**Soundness:** 2
**Presentation:** 2
**Contribution:** 2
**Rating:** 3
**Confidence:** 5

**Summary:**

The paper introduces Path Complex-based Message Passing (PCMP) and achieves promising results on molecular property prediction benchmarks. However, there are some major weakness in this paper including **lack of literature review, novelty and evaluation on current benchmarks**, and need to be further revised.

**Strengths:**

It proposes path complex-based message passing (PCMP) with some detailed graph theories, and achieves good results on some molecular tasks.

**Weaknesses:**

- Incorporate geometric features like bond, angles, dihedrals and improper angles in the modeling is not novel, as seen in Fig.2 of [1] and Fig.2 of [2]. There are also many works already involved these many-body systems.
- Although the paper introduces the concept of a “path complex,” the actual features used, which are bond distances and angles (Table 5), are standard. Similar methods already exist, such as hierarchical message passing for updating geometric embeddings in [2]. Also, models like NequIP, Allegro, and MACE employ **many-body expansions** in message passing, leveraging **tensor products** to incorporate higher-order geometric tensors (a path fusing process [3]), which are beyond the basic features mentioned in this paper. PCMP maybe a subset of the tensor products. A more comprehensive literature review is needed for the authors [4, 5].
- The primary weakness is that the authors claim that they are inspired by MD force fields, but **no results on any standard MD benchmark, such as MD17, rMD17, MD22 are provided**. Not to mention the conduction of MD simulations further driven by this MLFF. Since the paper focuses on molecular 3D structures, **it’s a necessity to proof invariance or equivariance**, which are missing here. In contrast, this paper provides a list of the graph path theories, appears more relevant to topological graphs and is insufficient to address geometric graphs. Furthermore, despite claiming that the method “enables systematic exploration of connectivity and interaction for analyzing complex systems and networks,” **there are no experiments on these tasks supporting this claim**.
- The GEM paper is relatively old and actually we do not need to generate 3D structures using RDKit from smiles for the 2D molecule datasets. Beside the datasets mentioned in the above question, **numerous 3D molecular datasets for DFT-level property prediction, such as QM9 (with 12 targets), OC20, OE62, and PCQM4Mv2, are available**. I strongly recommend evaluating PCMP on these benchmarks for a more comprehensive assessment.

[1] Wang, Yusong, et al. "Enhancing geometric representations for molecules with equivariant vector-scalar interactive message passing." Nature Communications 15.1 (2024): 313.

[2] Pei, Hongbin, et al. "Hago-net: Hierarchical geometric massage passing for molecular representation learning." Proceedings of the AAAI Conference on Artificial Intelligence. Vol. 38. No. 13. 2024.

[3] https://docs.e3nn.org/en/latest/api/o3/o3_tp.html

[4] Zhang, Xuan, et al. "Artificial intelligence for science in quantum, atomistic, and continuum systems." arXiv preprint arXiv:2307.08423 (2023). **Section 5.2**

[5] Han, Jiaqi, et al. "A Survey of Geometric Graph Neural Networks: Data Structures, Models and Applications." arXiv preprint arXiv:2403.00485 (2024).

**Questions:**

- missing ']' in Fig. 1 dihedral term.
- If the paper aims to emphasize the Path Weisfeiler-Lehman (PWL) capacity, it should evaluate the capacities of classical models, such as DimeNet, GemNet, and MACE. For reference, see the Geometric Weisfeiler-Lehman (WL) paper.

---

> ### Author Response · Authors · 2024-11-22
> **Response to Reviewer hgFA**
>
> Thank you for your careful review and valuable comments on our work.
>
> ## Weakness
> ### Weakness 1:  **_Many body expansions_**.
>
> Answer 1: While geometric features such as bond lengths, angles, dihedral angles, and improper angles are well-established in molecular modeling, previous studies often used both covalent and non-covalent bonds for angle calculations. **By focusing solely on covalent bonds, our method aligns more closely with real molecular structures. The natural ability of path complexes to represent these geometric features ensures their utility in molecular property prediction, as demonstrated in Table 5 (Page 10).**
>
> ### Weakness 2:  **_It’s a necessity to proof invariance or equivariance_**
>
> Answer2: **We have addressed this by adding Theorem 3.3 (Page 4), which proves the invariance of the path complex.** Our model achieves invariance by employing geometric features based on relative positional information (e.g., relative angles and distances), avoiding dependence on absolute coordinates. This ensures the model’s outputs remain unaffected by basic geometric transformations such as translations and rotations.
>
> ### Weakness 3: **_Despite claiming that the method “enables systematic exploration of connectivity and interaction for analyzing complex systems and networks,” there are no experiments on these tasks supporting this claim._**
>
> Answer3:  Thank you for pointing out the overreaching claim in our paper regarding the capability of our method to analyze complex systems and networks. **We acknowledge that our PCMP framework is primarily designed for data with rich geometric information, such as molecular structures, and currently lacks experimental validation for complex systems analysis.** We are actively working on adapting our model to better handle diverse data requirements, including non-geometric data. We will revise our manuscript to accurately reflect the scope and current capabilities of our research.
>
> ###Weakness 4: **_RDKit and 3D molecular datasets for DFT-level property prediction._**
>
> Answer4: We relied on RDKit because the original data files were in SMILES format. RDKit allowed us to generate atomic coordinates for extracting geometric features. Additionally, we compared our method against state-of-the-art models such as DMP【1】, SMPT【2】, and DGCL【3】. Our primary goal was to establish a novel geometric deep learning framework rather than simulate molecular dynamics, which informed our choice of datasets (e.g., QM9 for α, εHOMO, εLUMO predictions).
>
>
> ## Questions
> Answer:
> Thank you for your valuable suggestions. In our study, we have indeed recognized the importance of comparing our PCMP to well-established models. **Specifically, we have included comparisons of PCMP with DimeNet in our results section, particularly focusing on classification tasks for datasets such as Tox21, HIV, and MUV.** In these comparisons, PCMP demonstrated superior performance in terms of ROC-AUC metrics. We acknowledge the potential insights that could be gained from further comparisons with other models like GemNet and MACE.
>
>
> [1] Jinhua Zhu, Yingce Xia, Lijun Wu, Shufang Xie, Wengang Zhou, Tao Qin, Houqiang Li, and Tie-Yan Liu. 2023. Dual-view Molecular Pre-training. In Proceedings of the 29th ACM SIGKDD Conference on Knowledge Discovery and Data Mining (KDD '23). Association for Computing Machinery, New York, NY, USA, 3615–3627. https://doi.org/10.1145/3580305.3599317
>
> [2] Yishui Li, Wei Wang, Jie Liu, and Chengkun Wu. 2024. Pre-training molecular representation model with spatial geometry for property prediction. Comput. Biol. Chem. 109, C (Apr 2024). https://doi.org/10.1016/j.compbiolchem.2024.108023
>
> [3] Jiang X, Tan L, Zou Q. DGCL: dual-graph neural networks contrastive learning for molecular property prediction[J]. Briefings in Bioinformatics, 2024, 25(6): bbae474. https://academic.oup.com/bib/article/25/6/bbae474/7779242

---

> > ### Comment · Reviewer_hgFA · 2024-11-22
> >
> > **The authors’ rebuttal fails to convincingly address my concerns:**
> >
> > The authors acknowledge that many works already use geometric features such as bond lengths, angles, dihedral angles, and improper angles. Therefore, the novelty of this paper should lie in how these features are applied. However, the statement:
> >
> > *“By focusing solely on covalent bonds, our method aligns more closely with real molecular structures”*
> >
> > is fundamentally flawed. Interactions between atoms are not determined solely by covalent bonds but are also heavily influenced by **non-bonded interactions within a cutoff radius**. These non-bonded interactions are critical components of classical molecular mechanics (MM) force fields and play a central role in molecular dynamics (MD) simulations [1].
> >
> > $$
> > E_{MM} = E_{bond} + E_{non-bond}
> > $$
> >
> > If this work claims inspiration from MD, ignoring non-bonded interactions significantly undermines the credibility of the approach.
> >
> > [1] https://www.compchems.com/molecular-dynamics-non-bonded-interactions/#types-of-non-bonding-interactions
> >
> > The authors state that RDKit was used because the original data was in SMILES format and that RDKit helped generate atomic coordinates for extracting geometric features. However, this approach has notable limitations:
> >
> > Structures generated by RDKit are often unreliable for property calculations. Furthermore, RDKit frequently fails to generate valid 3D structures.  What we usually do is when it fails to generate valid 3D structures, fallback methods often resort to 2D representations (refer to GEM-2 [2] ).
> >
> > That is also the reason why I suggest the QM9 (with 12 targets), PCQM4Mv2 and other benchmarks in the initial review because they contain reasonable 3D structures with minimum relaxed energy.
> >
> > [2] https://github.com/PaddlePaddle/PaddleHelix/blob/dev/apps/pretrained_compound/ChemRL/GEM-2/pahelix/utils/compound_tools.py
> >
> > The authors claim:
> >
> > *“Our primary goal was to establish a novel geometric deep learning framework rather than simulate molecular dynamics, which informed our choice of datasets (e.g., QM9 for α, εHOMO, εLUMO predictions).”*
> >
> > This statement raises two issues:
> >
> > - **Ambiguity:**
> >
> > If the authors assert that their model, PCMP, can perform tasks related to QM9 (e.g., α, εHOMO, εLUMO), **no results or evaluations in the revision** are provided to support this.
> >
> > - **Inconsistency:**
> >
> > If the authors imply that QM9 and similar datasets are unsuitable for MD-related tasks, however, properties in QM9 like α, εHOMO, εLUMO are **quantum chemistry properties and unrelated to MD simulations.** There’s no apparent reason why PCMP cannot handle these tasks.

---

### Author Response · Authors · 2024-11-22
**General Response: The novelty of the proposed architecture**

# Novelty of the proposed architecture

Dear Reviewer,

Thank you for your careful review and valuable comments on our manuscript. In response to your concerns regarding the literature review, novelty, and evaluation on current benchmarks, we provide the following detailed explanations to better clarify the contributions and significance of our work.


**In this article, we draw inspiration from the fundamental principles of molecular dynamics. From a topological perspective, we utilize path complexes to encode the geometric features of molecular atomic coordinates, characterizing molecules. We combine this with geometric deep learning to implement message passing and pooling of path complexes, thereby enhancing the model’s performance in molecular property prediction tasks. Unlike direct simulations of molecular dynamics (MD) processes, our Path Complex Message Passing (PCMP) model inputs geometric features, not atomic coordinates, and thus cannot derive corresponding forces through differentiation. Therefore, our approach aligns more with topological deep learning rather than neural network potentials [1].**

1. Accurate Modeling of Molecular Force Field: Path complexes to naturally represent molecular bond angles and dihedral angles. For higher-order paths, such as 2-paths and 3-paths, we exclusively use covalent bonds to describe angles, dihedral angles, and other geometric features. **In our PCMP model, we exclusively use covalent bonds to construct higher-order paths (e.g., 2-paths and 3-paths), capturing angles and dihedral angles that better reflect many-body interactions in molecular force fields. The computational complexity of our algorithm is O(N^2), where N represents the number of atoms within a molecule.** Since our method only involves covalent bonds when constructing higher-order paths, it naturally limits the number of interactions that need to be processed, thus not excessively increasing the runtime.

2. Bidirectional Message Passing Mechanism: We proposed a universal bidirectional information transfer mechanism based on path complexes, which allows for the exchange of information between higher-order paths (such as 3-paths) and lower-order paths (such as 0-paths). This contrasts with traditional unidirectional information transfer (only from higher to lower order), significantly enhancing the model’s ability to capture the complexity of molecular structures, marking another innovation in our paper. Additionally, we demonstrated that the PWL test, based on the path complex. Through bidirectional message passing, we effectively aggregated information from both higher-order and lower-order paths, thereby enhancing predictive performance. Furthermore, we validated our approach on five benchmark datasets (two regression and three classification) to demonstrate its effectiveness and superiority.

3. Theoretical and Experimental Enhancements: **We added a new section (Theorem 3.3, Page 4) proving the invariance of the Path Complex. Furthermore, we have introduced new experimental results (Impact of Geometric Feature in Path Complex, Page 10) demonstrating that the inclusion of additional geometric features beyond bond and dihedral angles in higher-order paths significantly boosts model performance.**

Reference:

[1] Kocer E, Ko T W, Behler J. Neural network potentials: A concise overview of methods[J]. Annual review of physical chemistry, 2022, 73(1): 163-186.

---

> ### Comment · Reviewer_hgFA · 2024-11-22
>
> I have to comment this:
>
> **Inability to Derive Forces:**
>
> The authors state:
>
> *“Unlike direct simulations of molecular dynamics (MD) processes, our Path Complex Message Passing (PCMP) model inputs geometric features, not atomic coordinates, and thus cannot derive corresponding forces through differentiation. Therefore, our approach aligns more with topological deep learning rather than neural network potentials.”*
>
> However, **there are numerous works (e.g., DimeNet, DimeNet++, GemNet, PaiNN, TorchMD-Net, ViSNet) that incorporate geometric features and successfully derive forces through differentiation**. Here’s a straightforward method for achieving this:
>
> 1.	Enable gradients for atomic positions by setting requires_grad=True.
>
> 2.	Calculate geometric features (e.g., bond lengths, angles, etc.) based on these positions.
>
> 3.	Perform message passing and compute a scalar quantity, such as energy, as the final representation.
>
> 4.	Use torch.autograd.grad to compute the gradient of the energy with respect to the atomic positions. The negative of these gradients corresponds to the forces.
>
> Given this well-established approach, the inability to compute forces cannot be considered a valid limitation or excuse. Furthermore, if this paper is truly focused on topological deep learning rather than neural network potentials, why include tasks requiring 3D structures and do not compare with geometric graph neural networks (GNNs)? If the goal is purely topological modeling, tasks based on SMILES or other 2D representations would suffice.
>
> **Many-Body Interactions:**
>
> The authors claim:
>
> *“We exclusively use covalent bonds to construct higher-order paths (e.g., 2-paths and 3-paths), capturing angles and dihedral angles that better reflect many-body interactions in molecular force fields. The computational complexity of our algorithm is , where  represents the number of atoms within a molecule.”*
>
> However, many existing works (e.g., DimeNet, PaiNN) already capture angles, and others like GemNet and ViSNet handle dihedral angles explicitly. Additionally, models such as NequIP, Equiformer, and MACE employ many-body expansions that implicitly represent these features in a more theoretically grounded manner. **The claim that this method better reflects many-body interactions in molecular force fields is totally unsupported, especially since this model does not perform tasks related to molecular force fields.**
>
> **Computational Complexity:**
>
> Papers like PaiNN and MACE use density-based tricks to reduce the complexity of geometric feature calculations to $O(N)$. In contrast, this paper relies on explicit calculations, which are $O(N^2)$  and are already implemented in older models such as DimeNet and GemNet, which further use radial basis functions (RBF), angular basis functions (CBF), and spherical basis functions (SBF) to obtain smooth gradient. This approach does not introduce any novel computational efficiencies.
>
> **Conclusion:**
>
> I cannot identify the significant novelty in the approach described in this paper. The authors’ claims fail to provide sufficient evidence to substantiate their purported advantages and appear inconsistent with the paper’s stated goals.

---

> ### Author Response · Authors · 2024-11-23
> **Response to Reviewer hgFA’s Comments**
>
> We appreciate the reviewer pointing out the well-established methods for deriving forces via differentiation with respect to atomic coordinates. However, we would like to clarify that:
>
> ### On Computing Forces via Differentiation
>
> It is true that common force-predicting models rely on computing the gradient of the final predicted value with respect to input atomic coordinates to derive the forces on each atom. However, in our model, atomic coordinates are not used as direct inputs. This design choice provides strong invariance and equivariance properties, but it comes with the limitation that forces cannot be directly predicted. The reasons are as follows:
> 1. **Our model’s input variables include not only angles and dihedral angles but also higher-order geometric features, such as triangle areas formed by 2-path 3-body interactions and tetrahedron volumes formed by 3-path 4-body interactions. In this context, the predicted value  y  is a function of multiple variables beyond atomic coordinates.**
> 2. **These input variables span different scales, making it unreasonable to compute  y ’s gradient solely with respect to the coordinates  x . Such a calculation would ignore the contribution of higher-order geometric features to  y.**
>
> In this work, we focus on predicting global graph-level properties rather than atomic forces. **We acknowledge that force prediction could be achieved by modifying our PCMP framework to include atomic coordinates as inputs and adjusting the architecture accordingly.** This is a potential direction for future work.
>
> ### On the Rationality of Higher-Order Path Construction
>
> **Unlike previous models, we propose a novel framework for modeling higher-order information that is particularly natural for biomolecules. From a mathematical perspective, the elements in molecules—atoms, bonds, bond angles, and dihedral angles—correspond perfectly to 0-paths, 1-paths, 2-paths, and 3-paths, respectively.** Together, these paths form a path complex. To the best of our knowledge, this correspondence has not been explicitly explored in prior work. Highlighting this alignment was one of our primary motivations for this study.
>
> With the path complex representation of molecules, our framework offers several unique advantages:
> 1. **Direct higher-order modeling: Higher-order interactions can be explicitly modeled as higher-order paths.**
> 2. **Established mathematical foundation: Path complexes have a mature mathematical foundation, enabling the use of existing mathematical tools to study biomolecules.**
> 3. **Flexible message passing: The path complex representation allows for more flexible message-passing schemes, extending beyond pairwise interactions.**
>
> These three points distinguish our approach from existing models.
>
> ### Tasks Requiring 3D Structures
>
> We included tasks requiring 3D structures as a means to validate the ability of our path complex framework to capture geometric information, which is intrinsic to many molecular properties. This does not imply an intention to replicate or compete with geometric GNNs. Instead, we aim to demonstrate the potential of path complexes to provide a new perspective on molecular representations.
>
> We agree with the reviewer that topological tasks based on 2D representations (e.g., SMILES or molecular graphs) would also be relevant, and we plan to explore these tasks in future work.
>
> ### Computational Complexity
>
> We acknowledge the reviewer’s comment regarding computational complexity. While some models (e.g., PaiNN, MACE) utilize density-based techniques to achieve \( O(n) \) complexity, our approach prioritizes the explicit calculation of geometric features to preserve the interpretability and flexibility of representing higher-order paths.
>
> Additionally, while radial basis functions (RBF) and spherical basis functions (SBF) are widely used to smooth gradients, they are orthogonal to the contributions of our path complex framework. **Our work does not aim to introduce novel computational efficiencies but instead focuses on providing a new mathematical perspective for modeling molecular systems.ng a new perspective for modeling molecular systems.**
>
> ### Conclusion
>
> **Our goal is not to develop a new neural network potential method. Instead, we aim to integrate path complexes with molecular force fields from a mathematical and physical perspective, providing a novel framework for predicting biomolecular properties.**
> We acknowledge the limitations of our current approach and view them as opportunities for future improvement. We sincerely thank the reviewer for their detailed comments and will consider incorporating additional tasks and comparisons to further validate the applicability of our framework

---

> ### Comment · Reviewer_hgFA · 2024-11-23
>
> 1.	**On Higher-Order Geometric Features:**
>
> The authors state:
>
> *“Our model’s input variables include not only angles and dihedral angles but also higher-order geometric features, such as triangle areas formed by 2-path 3-body interactions and tetrahedron volumes formed by 3-path 4-body interactions.”*
>
> However, numerous previous works, from DimeNet and GemNet to NequIP and MACE, already incorporate higher-order geometric tensors beyond 3-body interactions. **These methods utilize similar or even more advanced geometric representations while maintaining their ability to predict molecular properties and forces.** The claim that these features are unique to this work is unsupported and overlooks the substantial body of prior research.
>
> 2.	**On Gradient Calculation and Force Prediction:**
>
> The authors argue:
>
> *“It’s unreasonable to compute ’s gradient solely with respect to the coordinates x. Such a calculation would ignore the contribution of higher-order geometric features to y.”*
>
> This statement is totally problematic and misleading. The computation of forces as gradients of the energy with respect to atomic coordinates is a cornerstone of modern machine learning force fields. Models such as DimeNet, GemNet, NequIP, and MACE explicitly incorporate many-body interactions and high-order geometric tensors, **which mostly contribute to the accurate prediction on both molecular properties and forces.**  **Suggesting that taking gradient on force prediction is “unreasonable” undermines the validity of these well-established works and lacks theoretical justification.**
>
> 3.	**On Benchmarking and Comparisons:**
>
> The authors state:
>
> *“We included tasks requiring 3D structures as a means to validate the ability of our path complex framework to capture geometric information, which is intrinsic to many molecular properties. This does not imply an intention to replicate or compete with geometric GNNs.”*
>
> If the focus is on validating the ability to capture geometric information, comparisons with state-of-the-art geometric GNNs such as DimeNet, PaiNN, GemNet, NequIP, and MACE are mandatory. Without such comparisons, conducting 3D molecular tasks is effectively meaningless. Furthermore:
>
> - If the work is truly aimed at topological deep learning, there should be evaluations on established topological benchmarks.
> - If the goal is to work on 3D molecular structures, comparisons with prevailing benchmarks and baselines in this domain are essential. The lack of performance on either front is a major limitation.
>
> 4.	**On Broader Predictions:**
>
> **Even if force prediction is beyond the scope of this work, the model could still address molecular property prediction tasks such as energy, HOMO, and LUMO (e.g., benchmarks in QM9).** The absence of performance evaluation on such standard tasks leaves the contributions of this paper unsubstantiated.
>
> **Conclusion:**
>
> All the aforementioned works (DimeNet, GemNet, NequIP, MACE, etc.):
>
> 1.	Involve many-body interactions.
>
> 2.	Utilize high-order geometric tensors.
>
> 3.	Predict both molecular properties and forces.
>
> **I strongly recommend the authors carefully study these papers in this field before making unsupported claims. You still have time to conduct several experiments and comparison as other reviewers also suggested during the rebuttal period** Moreover, failing to compare against geometric GNNs on molecular 3D tasks or against topological deep learning methods on topological benchmarks raises serious questions about the completeness of this work.

---

### Meta-Review · Area_Chair_Q1P7 · 2024-12-22

**Metareview:**

Majority of the reviewers raised significant concerns on this paper and the overall support is very weak.

**Additional Comments On Reviewer Discussion:**

There have been extensive discussions, but some of the major issues are not resolved.

---

### Decision · Program_Chairs · 2025-01-22

Reject